# PhySense: Sensor Placement Optimization for Accurate Physics Sensing

**Yuezhou Ma, Haixu Wu, Hang Zhou, Huikun Weng, Jianmin Wang, Mingsheng Long**[✉]
School of Software, BNRist, Tsinghua University, China
{mayuezhou20,wuhaixu98}@gmail.com, {zhou-h23,wenghk22}@mails.tsinghua.edu.cn,
{jimwang,mingsheng}@tsinghua.edu.cn

## Abstract

Physics sensing plays a central role in many scientific and engineering domains, which inherently involves two coupled tasks: reconstructing dense physical fields from sparse observations and optimizing scattered sensor placements to observe maximum information. While deep learning has made rapid advances in sparse-data reconstruction, existing methods generally omit optimization of sensor placements, leaving the mutual enhancement between reconstruction and placement on the shelf. To change this suboptimal practice, we propose PhySense, a synergistic two-stage framework that learns to jointly reconstruct physical fields and to optimize sensor placements, both aiming for accurate physics sensing. The first stage involves a flow-based generative model enhanced by cross-attention to adaptively fuse sparse observations. Leveraging the reconstruction feedback, the second stage performs sensor placement via projected gradient descent to satisfy spatial constraints. We further prove that the learning objectives of the two stages are consistent with classical variance-minimization principles, providing theoretical guarantees. Extensive experiments across three challenging benchmarks, especially a 3D geometry dataset, indicate PhySense achieves state-of-the-art physics sensing accuracy and discovers informative sensor placements previously unconsidered. Code is available at this repository: https://github.com/thuml/PhySense.

## 1 Introduction

Physics sensing remains a foundational task across several domains [45, 23, 47], including fluid dynamics [32, 15, 50], meteorology [39, 5], and industrial applications [13, 3]. The task aims to reconstruct the spatiotemporal information of a physical system from limited sparse observations gathered by pre-placed sensors. In real-world scenarios, the number of sensors is usually limited due to spatial constraints, power consumption, and environmental restrictions. As a result, allocating the limited sensors to the most informative positions is crucial for high reconstruction accuracy. Poorly placed sensors lead to spatial blind spots and information loss, resulting in insufficient observations, while well-placed sensors enable the recovery of fine-scale physical patterns. Therefore, advancing accurate physics sensing requires not only improving the capabilities of reconstruction models, but also developing principled strategies to determine *optimal sensor placements*.

Recent advances in deep learning have demonstrated remarkable capabilities in sparse-data reconstruction, owing to its ability to approximate highly nonlinear mappings [16, 44, 37] between scattered observations and dense physical fields. Current reconstruction methods fall into two categories: deterministic and generative. Deterministic methods such as VoronoiCNN [14] employ Voronoi-tessellated grids for CNN-based reconstruction, while Senseiver [41] utilizes implicit neural representations [19] to establish correlations between sparse measurements and query points. Moreover, due to the inherently ill-posed nature [22] of the reconstruction task, where sparse observations provide insufficient

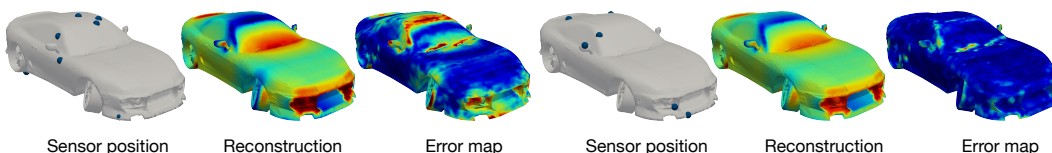

Figure 1: Performance comparison under same reconstruction model but different sensor placements. (a) Random placement yields poor results due to inadequate spatial coverage. (b) Our optimized placement achieves accurate reconstruction by discovering informative regions, including side mirrors.

information for recovering dense fields, generative models [17, 42, 30, 29] have been increasingly explored. For example, DiffusionPDE [18] and $S^3GM$ [28] model physical field distributions via diffusion models, and integrate sensor observations during sampling through training-free guidance [7, 49]. Despite their reconstruction accuracy, these methods share a fundamental limitation: in their practical experiments, sensor placements are either randomly distributed or arbitrarily fixed rather than being systematically optimized. Consequently, the synergistic potential of co-optimizing reconstruction quality and placement strategy remains largely unexplored.

In practice, sensor placement determines the structure of the observation space and constrains what the model can perceive, while the feedback from model's reconstruction should guide where sensing is most beneficial. Notably, as illustrated in Figure 1, even under the identical reconstruction model, the performance varies dramatically with different sensor placements. Random sensor placement fails to capture important spatial regions, resulting in degraded performance, whereas placements strategically optimized through reconstruction feedback yield significantly enhanced reconstruction accuracy by targeting informative regions, such as side mirrors of a car. These results suggest the existence of a positive feedback loop between reconstruction quality and sensor placement optimization.

Building upon these insights, this paper presents PhySense, a synergistic two-stage framework that combines physical field reconstruction with sensor placement optimization. In the first stage, we train a base reconstruction model using flow matching [29] capable of generating dense fields from all feasible sensor placements. In the second stage, leveraging the reconstruction feedback, sensors are optimized using *projected gradient descent* to satisfy spatial geometries. Furthermore, we prove that the learning objectives of the two stages are consistent with classical variance-minimization principles, providing theoretical guarantees. We evaluate our method across three challenging benchmarks, including turbulent flow simulations, reanalysis of global sea temperature, and industrial simulations of aerodynamic surface pressure over a 3D car on an irregular geometry. In all cases, PhySense consistently achieves state-of-the-art performance and discovers some informative sensor placements previously unconsidered. Overall, our contributions can be summarized as follows:

- We introduce PhySense, a synergistic two-stage framework for accurate physics sensing, which integrates a flow-based generative reconstruction model with a sensor placement optimization strategy through *projected gradient descent* to respect spatial constraints.
- We prove the learning objectives of reconstruction model and sensor placement optimization are consistent with classical variance-minimization targets, providing theoretical guarantees.
- PhySense achieves consistent state-of-the-art reconstruction accuracy with 49% relative gain across three challenging benchmarks and discovers informative sensor placements.

## 2 Preliminaries

### 2.1 Deep Reconstruction Models

Recent deep learning methods have demonstrated promising performance in reconstructing dense physical fields from sparse measurements. Existing reconstruction models can be broadly categorized into deterministic and generative approaches. Representative deterministic methods are VoronoiCNN [14] and Senseiver [41]. VoronoiCNN constructs a Voronoi tessellation from scattered sensor observations, and learns to map the resulting structured representation to the target physical field. Senseiver encodes arbitrarily sized scattered inputs into a latent space using cross-attention and adopts an implicit neural representation as its decoder. However, since sparse reconstruction is an

inherently ill-posed problem, these deterministic methods lack the ability to characterize uncertainty in the reconstructed fields. Deep generative models have also been applied to the sparse reconstruction task. For example, S$^3$GM [28] and DiffusionPDE [18] both tackle the reconstruction task by first learning a generative model of the underlying dynamics and then guiding the sampling process using sparse sensor observations. Crucially, these methods only utilize sparse observations during the sampling process—not during training—thus requiring thousands of sampling steps to align with sparse inputs, which compromises the efficiency of deep models. Despite promising reconstruction accuracy, all their experiments are conducted under random or fixed sensor placements, which leads to suboptimal reconstruction accuracy due to potentially non-informative sensor placements.

## 2.2 Sensor Placement Optimization

**Optimal criterion** Optimal sensor placement aims to maximize information extraction from physical systems by strategically positioning sensors. This problem is rooted in optimal experimental design (OED) theory [33, 36], which provides basic criteria for evaluating sensor placements. Three most widely-used criteria are: (1) *A-optimality*, which minimizes the average variance of all data estimates; (2) *D-optimality*, which maximizes the overall information content; and (3) *E-optimality*, which focuses on the worst-case estimation error. More details about these criteria can be found in [32]. In this work, we primarily focus on *A-optimality*. While these criteria originated in classical OED, they have been adapted to sensor placement optimization by incorporating spatial constraints [26].

**Optimization algorithm** Sensor placement optimization is a fundamentally hard problem due to its inherent high-dimensional nature. For finite option problems, an exhaustive search over all feasible placements may be tractable, but for more common high-dimensional problems, principled approximations become necessary. The theories of compressed sensing [9] show that signals admitting sparse representations in a known basis, such as the Fourier basis, can be accurately reconstructed with a small number of sensor observations. However, in real-world applications, the assumption of a suitable and known basis may not hold. Moreover, recent advances in machine learning have enabled data-driven discovery of low-dimensional structures such as low-rank subspaces [11, 46, 27]. This shift from fixed to data-driven representation spaces has spurred the development of placement optimization methods that leverage empirical priors or learned latent bases for efficient compression and reconstruction [32, 40]. For example, the SSPOR algorithm [32] first learns a data-driven basis, then applies QR factorization to select basis-specific sensor locations that maximize reconstruction performance. While recent work has begun to explore statistical-based sensor placement, deep learning-based approaches remain under explored. Besides, the further challenge lies in how to effectively co-optimize the reconstruction and the placement for accurate physics sensing.

## 2.3 Flow-based Models

Flow matching [29, 30, 31, 1] is a framework for solving transport mapping problem: given two distributions $\mathbf{X}_0 \sim \pi_0$ and $\mathbf{X}_1 \sim \pi_1$, the goal is to find a continuous mapping $T$, such that $T(\mathbf{X}_0) \sim \pi_1$. The method's efficiency and capacity have made it particularly valuable in vision tasks [12]. Unlike diffusion models that require computationally intensive ODE [42] or SDE [17, 43, 24] solving, flow matching introduces an ODE model that transfers $\pi_0$ to $\pi_1$ optimally *via the straight line*, theoretically the shortest path between two points, $\mathrm{d}\mathbf{X}_t = \mathbf{v}_\theta(\mathbf{X}_t, t)\,\mathrm{d}t, t \in [0, 1]$, where $\mathbf{v}_\theta(\mathbf{X}_t, t)$ is a learnable velocity field. To estimate $\mathbf{v}$, flow matching solves a simple least squares regression problem that fits $\mathbf{v}_\theta$ to $(\mathbf{X}_1 - \mathbf{X}_0)$. The resulting dynamics generate samples via efficient push-forward along the theoretically shortest path, $\mathbf{X}_{t+\mathrm{d}t} = \mathbf{X}_t + \mathbf{v}_\theta(\mathbf{X}_t, t)\mathrm{d}t$, where $\mathbf{X}_0 \sim \pi_0$. In this paper, we consider to integrate this advanced technique for a more efficient reconstruction model.

## 3 Method

To enable accurate sensing of physical fields, we co-optimize reconstruction and placement through a synergistic two-stage framework. Our approach begins by training a flow-based model capable of reconstructing dense fields from scattered measurements across all feasible placements. Leveraging the reconstruction feedback, PhySense seamlessly integrates reconstruction and placement processes, forming a closed loop (Fig. 2 (a)), and formulates placement as a spatially constrained optimization problem, efficiently solved via projected gradient descent. Moreover, the optimization objective theoretically provides a bidirectional control with respect to the variance-minimization target.

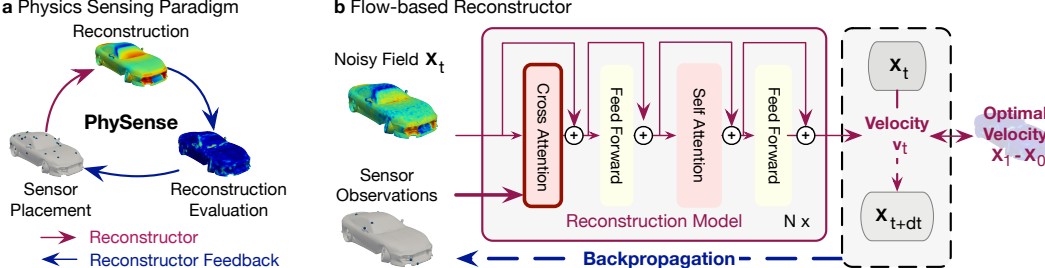

Figure 2: (a) PhySense works as a closed-loop physics sensing paradigm that iteratively co-optimizes sensor placements and reconstruction quality under the reconstruction feedback. (b) Reconstruction stage: A flow-based model with cross-attention mechanisms trained to process arbitrary sensor placements ('←--' indicates information flow for subsequent generation and placement optimization).

**Problem setup**   Let $\mathcal{D} \subset \mathbb{R}^d$ be a bounded domain, where exists $\mathcal{C}$ physical fields of interest. We consider $\mathcal{M} = \{p_1, \ldots, p_m\}$ sensors distributed across $\mathcal{D}$, where each sensor measures the target quantities at its position. Our *physics sensing task* is to accurately reconstruct the entire physical field $\mathbf{X}_1$ based on sparse sensor observations $\mathbf{X}_1(\mathbf{p}^*)$ under the optimal sensor placement $\mathbf{p}^*$.

## 3.1   Flow-based Reconstructor

Physics sensing, as an inverse problem to reconstruct dense physical fields from sparse observations, naturally requires generative approaches capable of handling the inherent ill-posedness. In our framework, we formulate the reconstruction task as an optimal transport flow problem between the target physical field distribution $\pi_1$ and a standard Gaussian source distribution $\pi_0$, with sensor placements and sparse observations serving as *conditioning variables*.

Notably, the reconstruction model is expected to provide instructive feedback for any current sensor placement in the subsequent placement optimization stage. Therefore, we implement random sensor placement sampling during the flow model training, which helps the model extract maximal useful information from any feasible sensor placement. Therefore, our training flow loss is formalized as

$$\mathcal{L} = \int_0^1 \mathbb{E}_{\mathbf{X}_0, \mathbf{X}_1, \mathbf{p}} \left[ \|(\mathbf{X}_1 - \mathbf{X}_0) - \mathbf{v}_\theta(\mathbf{X}_t, t, \mathbf{p}, \mathbf{X}_1(\mathbf{p}))\|^2 \right] \mathrm{d}t, \quad \mathbf{X}_t = t\mathbf{X}_1 + (1-t)\mathbf{X}_0, \quad (1)$$

where $\mathbf{v}_\theta$ represents our learned velocity model, and $\mathbf{p}$ denotes a sensor placement randomly sampled from a uniform distribution on $\mathcal{D}$. This model design must address two practical challenges: (1) handling diverse physical domains spanning regular grids to irregular meshes, and (2) effectively incorporating sparse observations for consistent reconstruction. For the first challenge, we leverage the state-of-the-art DiT [35, 10] for regular grids and Transolver [48], the most advanced approach for general geometry modeling, for irregular geometries, ensuring optimal parameterization of the velocity field across various domains. The second challenge is resolved through a cross-attention mechanism that dynamically associates scattered sensor measurements with their relevant spatial influence regions during training. This not only provides some physical insights into sensor-field interactions but also enhances generalization across varying number and placements of sensors.

Theoretically, we prove that our reconstructor can learn the conditional expectation of physical fields.

**Theorem 3.1** (**Flow-based reconstructor is an unbiased estimator of physical fields**). *Let the training flow loss Eq.* (1) *be minimized over a class of velocity fields* $\mathbf{v}$. *The **optimal learned velocity**,*

$$\mathbf{v}^*(x, t, \mathbf{p}, \mathbf{X}_1(\mathbf{p})) = \mathbb{E}[\mathbf{X}_1 - \mathbf{X}_0 \mid \mathbf{X}_t, t, \mathbf{p}, \mathbf{X}_1(\mathbf{p})]$$

*equals the conditional mean of all feasible directions between the target data* $\mathbf{X}_1$ *and the initial noise* $\mathbf{X}_0$. *Define the reconstructed data as the integral along the optimal flow,* $\bar{\mathbf{X}}_1 := \mathbf{X}_0 + \int_0^1 \mathbf{v}^*(\mathbf{X}_t, t, \mathbf{p}, \mathbf{X}_1(\mathbf{p})) \mathrm{d}t$, *then* $\bar{\mathbf{X}}_1$ *is an unbiased estimator of the target physical fields given sensor placement* $\mathbf{p}$ *and corresponding observation* $\mathbf{X}_1(\mathbf{p})$, *i.e.* $\mathbb{E}[\bar{\mathbf{X}}_1 \mid \mathbf{p}, \mathbf{X}_1(\mathbf{p})] = \mathbb{E}[\mathbf{X}_1 \mid \mathbf{p}, \mathbf{X}_1(\mathbf{p})]$.

## 3.2   Sensor Placement Optimization

Based on the well-trained reconstructor, we formalize placement optimization as a constrained optimization problem efficiently solved through a newly proposed projected gradient descent strategy to respect spatial constraints, yielding near A-optimal [33] placements with theoretical guarantees.

**Projected gradient descent** Optimizing sensor placement presents unique challenges compared to standard model parameter optimization, since solutions must adhere to complex spatial constraints imposed by the physical domain $\mathcal{D}$. While gradient-based updates can suggest improved placements, naive application often violates geometric feasibility—particularly in irregular geometries. Therefore, we employ *projected gradient descent* (Fig. 3), which iteratively (1) computes accuracy-improving gradients and (2) projects the updated placements back to the feasible space via nearest-point mapping. This process can be formalized as

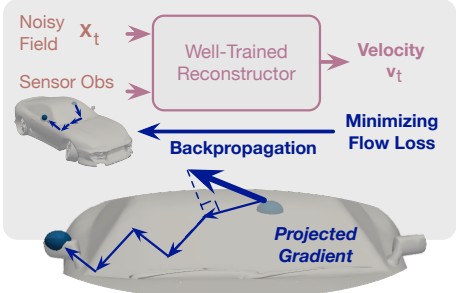

Figure 3: Placement optimization stage: Based on a well-trained reconstructor, sensors are optimized by geometry-constrained projected gradient descent to minimize the flow loss with theoretical guarantees.

$$\mathbf{p}^{(k+1)} = \text{Proj}_{\mathcal{D}}\left(\mathbf{p}^{(k)} - \eta\nabla_{\mathbf{p}}\mathcal{L}_{\text{flow}}(\mathbf{p}^{(k)})\right), \quad (2)$$

where $\mathbf{p}^{(k)}$ denotes the sensor placement at the $k$-th iteration, and $\text{Proj}_{\mathcal{D}}(\cdot)$ denotes the projection operator onto the feasible domain $\mathcal{D}$. The strategy maintains strict spatial constraints while systematically exploring superior placements, effectively bridging unconstrained optimization with physical feasibility requirements.

**Optimization objective** The selection of optimization objective significantly impacts both computational cost and reconstruction accuracy. Although minimizing the reconstruction error $\|\bar{\mathbf{X}}_1 - \mathbf{X}_1\|^2$ seems conceptually straightforward, its requirement for backpropagation among multiple flow-process steps results in prohibitive computational and memory overhead. In contrast, the flow loss defined in Eq. (1) maintains computational tractability through single flow-process step gradient calculation and, due to the optimal transport path between $\mathbf{X}_0$ and $\mathbf{X}_1$, theoretically guarantees near A-optimal sensor placements, a dual advantage that makes it our preferred choice.

The following theoretical analysis elucidates what our placement optimization objective fundamentally captures and how it relates to the well-established A-optimality framework.

**Assumption 3.2** (**Conditional Gaussian assumption**). *We assume the target distribution is conditionally Gaussian, specifically,* $\mathbf{X}_1 \mid (\mathbf{p}, \mathbf{X}_1(\mathbf{p})) \sim \mathcal{N}(\mathbf{0}, \Sigma_{\mathbf{p}})$*, where* $\Sigma_{\mathbf{p}}$ *is the covariance matrix determined by the sensor placement* $\mathbf{p}$*. Furthermore,* $\mathbf{X}_0$ *and* $\mathbf{X}_1$ *are mutually independent.*

**Definition 3.3** (**Classical A-optimal placement**). *We define the conditional covariance matrix of* $\mathbf{X}_1$ *given placement* $\mathbf{p}$ *and corresponding observation* $\mathbf{X}_1(\mathbf{p})$ *is* $\text{Var}(\mathbf{p}) = \mathbb{E}[(\mathbf{X}_1 - \mathbb{E}\mathbf{X}_1)(\mathbf{X}_1 - \mathbb{E}\mathbf{X}_1)^{\mathsf{T}} \mid \mathbf{p}, \mathbf{X}_1(\mathbf{p})] = \Sigma_{\mathbf{p}}$*. A sensor placement* $\mathbf{p}$ *is called* **A–optimal** *if it minimizes the trace of* $\Sigma_{\mathbf{p}}$*,*

$$\mathbf{p}^{A\text{-}opt} := \arg\min_{\mathbf{p}\in\mathcal{D}^m} \mathcal{L}_{\text{A}}(\mathbf{p}), \quad \text{with } \mathcal{L}_{\text{A}}(\mathbf{p}) := \text{tr}(\Sigma_{\mathbf{p}}) \equiv \mathbb{E}\|\mathbf{X}_1 - \mathbb{E}\mathbf{X}_1\|^2,$$

*where* $\mathcal{D}^m$ *denotes all feasible placements for* $m$ *sensors. From definition, an A–optimal placement attains the minimal total conditional variance of the reconstructed field among feasible placements.*

**Definition 3.4** (**Flow-loss-minimized sensor placement**). *Assume the velocity model* $\mathbf{v}$ *attains its optimum* $\mathbf{v}^*$*. Another optimal placement is defined if it minimizes the residual flow loss* $\mathbf{p}^* = \arg\min_{\mathbf{p}\in\mathcal{D}^m} \mathcal{L}_{\text{flow}}(\mathbf{p})$*, where* $\mathcal{L}_{\text{flow}} := \mathbb{E}_{\mathbf{X}_t,t}\left[\left\|(\mathbf{X}_1 - \mathbf{X}_0) - \mathbf{v}^*\right\|^2\right]$*, which measures the conditional variance when the optimal transport flow path is used instead of the true data transition.*

**Theorem 3.5** (**The objectives of two optimal placements are mutually controlled**). *Under Assumption 3.2, the objectives of two optimal placements can be simplified to*

$$\mathcal{L}_{\text{A}}(\mathbf{p}) = \sum_{i=1}^{d}\lambda_i(\mathbf{p}), \ \mathcal{L}_{\text{flow}}(\mathbf{p}) = \sum_{i=1}^{d}\frac{\pi}{2}\sqrt{(\lambda_i(\mathbf{p}))} \quad (3)$$

*with eigenvalues* $\{\lambda_i(\mathbf{p})\}_{i=1}^{d}$ *of* $\Sigma_{\mathbf{p}}$*. Thus, two objectives control each other via following inequality:*

$$\frac{2}{\pi^2}\mathcal{L}_{\text{flow}}^2(p) \le \mathcal{L}_{\text{A}}(p) \le \frac{4}{\pi^2}\mathcal{L}_{\text{flow}}^2(p). \quad (4)$$

*Proof sketch.* The proof's core lies in the non-trivial simplification of $\mathcal{L}_{\text{flow}}$ through careful analysis of the structure of the conditional covariance matrix. Owing to the favorable properties of the optimal transport path, the involved complex integrals can be calculated in a closed form, offering further insight into the structure of the flow matching. More details can be found in Appendix A.2. $\square$

**Remark 3.6.** *Consequently, the two objectives admit a two-sided polynomial bound of degree 2, which implies that minimizing $\mathcal{L}_{\text{flow}}$ inherently constrains $\mathcal{L}_{A}$ to lie within a bounded neighborhood of its optimum, and vice versa. This provides a theoretical guarantee for achieving **near A-optimal sensor placement** under the efficiently implementable flow-loss-minimization objective.*

**Overall design** In summary, PhySense seamlessly integrates two synergistic stages for accurate physics sensing. (1) Reconstruction: a flow-based model learns to reconstruct dense physical fields from arbitrary scattered inputs through optimal transport flow dynamics. During training, both the number and placements of sensors are randomized to ensure generalizability. (2) Placement optimization: under fixed-number sensors, placements are optimized on the residual flow loss via projected gradient descent, constrained to feasible geometries (e.g., 3D surfaces). Theoretically, we prove that the reconstruction model learns the conditional expectation of target physical fields, and the flow-loss minimization objective polynomially controls the classical A-optimal criteria.

# 4 Experiments

We perform comprehensive experiments to evaluate PhySense, including turbulent flow simulation, reanalysis of global sea temperature with land constraints and aerodynamic pressure on a 3D car.

**Benchmarks** As presented in Table 1, our experiments encompass both regular grids and unstructured meshes in the 2D and 3D domains. The benchmarks include: (1) Turbulent Flow: numerical simulations of channel flow turbulence following the setup in Senseiver [41]; (2) Sea Temperature: 27-year daily sea surface temperature at $1°$ resolution

Table 1: Summary of benchmarks in PhySense. #Mesh records the size of discretized mesh points.

| Benchmarks | Geometry | #Dim | #Mesh |
|---|---|---|---|
| Turbulent Flow | Regular Grid | 2D | 6144 |
| Sea Temperature | Land Constraints | 2D | 64800 |
| Car Aerodynamics | Unstructured Mesh | 3D | 95428 |

with intricate land constraints from the GLORYS12 reanalysis dataset [21]; and (3) Car Aerodynamics: high-fidelity OpenFOAM simulations [20] of surface pressure over a complex car geometry from ShapeNet [6], where we reconstruct the irregular pressure fields from scattered sensor observations.

**Baselines** We evaluate PhySense via two approaches: (1) comparing the reconstruction model under randomly sampled sensor placements, and (2) comparing sensor placements with frozen PhySense's reconstruction model. The reconstruction benchmarks include five representative baselines: classical statistical-based method SSPOR [32], deterministic models VoronoiCNN [14] and Senseiver [41], and generative approaches $S^3$GM [28] and DiffusionPDE [18]. For placement evaluation, we compare our optimized placement against random sampling and SSPOR's optimized placement [32].

**Implementations** All methods of each benchmark maintain identical training configurations including epochs, batch sizes, ADAM optimizer [25], and maximum learning rate, while preserving other details as specified in their original papers. For the reconstruction model, we employ DiT [35] for turbulent flow and sea temperature benchmarks, while using Transolver [48] as a Transformer backbone to fit irregular mesh for car aerodynamics benchmark. The placement optimization stage runs for 5 epochs with the frozen reconstruction model. Both two stages employ relative L2 as the metrics for the reconstructed fields. Additional implementation details are provided in Appendix C.

## 4.1 Turbulent Flow

**Setups** The benchmark focuses on reconstructing turbulent flow within a channel using a 2-D slice discretized on a $128 \times 48$ regular grid. This dataset is widely used, with classical models such as Senseiver [41] and VoroniCNN [14] serving as benchmarks. We replicate Senseiver's setup; specifically, 25–300 sensors are randomly selected to train the reconstructor, with sensor positions uniformly sampled across the entire domain. A setting with 30 sensors represents low coverage, while 200 sensors corresponds to high coverage. The model is trained using all available simulation data and evaluated on its ability to reconstruct the corresponding fields under varying sensor placements. Besides, due to the computational cost of DiffusionPDE and $S^3$GM—each reconstruction involving thousands of model evaluations—we restrict their evaluation to 500 randomly sampled cases.

**Results** As shown in Table 2, PhySense achieves state-of-the-art performance among all sensor counts, surpassing the second-best method by a substantial margin. Despite starting from our strong

Table 2: Performance comparison on the turbulent flow and sea surface temperature benchmarks. Relative L2 error is reported. *PhySense-opt* denotes the performance under optimized sensor placement, while all other deep models, including *PhySense*, use randomly placed sensors. Besides, SSPOR is a classical method that performs both placement optimization and reconstruction. # indicates the sensor number. Promotion refers to the relative error reduction of PhySense-opt w.r.t. the best baseline.

| Models | Placement Strategy | Turbulent Flow | | | | Sea Temperature | | | |
|---|---|---|---|---|---|---|---|---|---|
| | | #200 | #100 | #50 | #30 | #100 | #50 | #25 | #15 |
| SSPOR [32] | SSPOR | 0.3018 | 0.4429 | 0.5921 | 0.6637 | 0.0756 | 0.0752 | 0.0789 | 0.0719 |
| VoronoiCNN [14] | | 0.3588 | 0.4870 | 0.6919 | 0.7992 | 0.1496 | 0.1537 | 0.2433 | 0.2817 |
| Senseiver [41] | Random | 0.1842 | 0.2316 | 0.3740 | 0.5746 | 0.0715 | 0.0732 | 0.0769 | 0.0784 |
| S$^3$GM [28] | | 0.1856 | 0.2016 | 0.2681 | 0.6421 | 0.1115 | 0.1137 | 0.1581 | 0.1795 |
| DiffusionPDE [18] | | 0.1993 | 0.3312 | 0.6604 | 0.9163 | 0.0664 | 0.0775 | 0.0859 | 0.1139 |
| **PhySense** | Random | **0.1233** | **0.1527** | **0.2586** | **0.5176** | **0.0439** | **0.0452** | **0.0477** | **0.0520** |
| **PhySense-opt** | Optimized | **0.1106** | **0.1257** | **0.1558** | **0.2157** | **0.0426** | **0.0430** | **0.0437** | **0.0439** |
| Relative Promotion | | 39.96% | 38.56% | 41.89% | 62.46% | 35.84% | 41.26% | 43.17% | 38.94% |

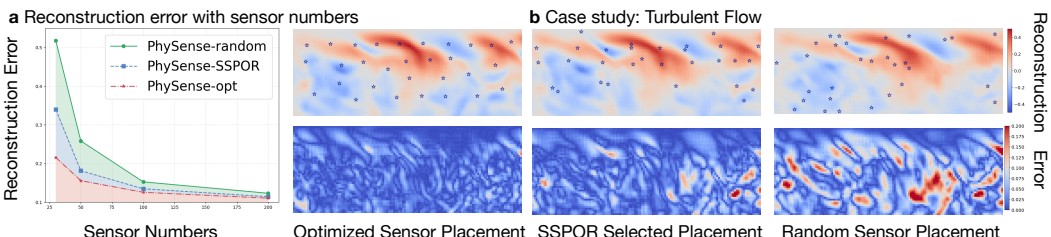

Figure 4: (a) Reconstruction loss versus the number of sensors. Random sampling, classical SSPOR, and our optimized method are compared using the same PhySense reconstruction base model. (b) Visualization of reconstruction results, error maps, and sensor placements (denoted by ⋆). Our optimized placement clearly outperforms the other two strategies on turbulent flow benchmark.

reconstruction model, the reconstruction accuracy is further significantly improved through sensor placement optimization. The improvement is particularly notable under low coverage with 30 sensors relative to the high coverage setup with 200 sensors. Remarkably, even with fewer sensors, the optimized 30-sensor placement exceeds the performance of the unoptimized 50-sensor counterpart.

As shown in Fig. 4(a), our learned placement strategy consistently yields the lowest reconstruction error compared to the other two strategies, especially in low coverage settings, demonstrating its superior effectiveness. Fig. 4(b) further supports this observation: without any explicit spatial constraints, the learned sensors are distributed in a way that avoids excessive concentration, thereby reducing redundancy and enhancing information gain under the same number of sensors.

## 4.2 Sea Temperature

**Setups** This benchmark is derived from the GLORYS12 reanalysis dataset [21], focusing on the sea surface temperature variable. The data are downsampled to a $1° × 1°$ resolution, resulting in global ocean fields of size $360 × 180$. During training, between 10 and 100 sensors are randomly selected, with all sensors uniformly sampled from valid ocean regions, excluding land areas. The reconstruction task aims to recover dense temperature fields from these sparse observations. Our training set comprises 9,843 daily samples from 1993 to 2019, with data from 2020–2021 reserved for testing. This setup allows us to evaluate two critical aspects of model capability: (1) adaptability to varying sensor placements, and (2) generalization to temporal distribution shifts. Moreover, the presence of complex land–sea boundaries introduces significant challenges for both reconstruction and sensor placement optimization. We address this by providing a land mask as an additional input.

**Results** PhySense-opt consistently achieves state-of-the-art performance across all sensor numbers (average improvement 40.0%), as shown in Table 2. Besides, SSPOR [32] performs reasonably well on this benchmark, due to the relatively stable temporal variations of sea temperature over the 30-year

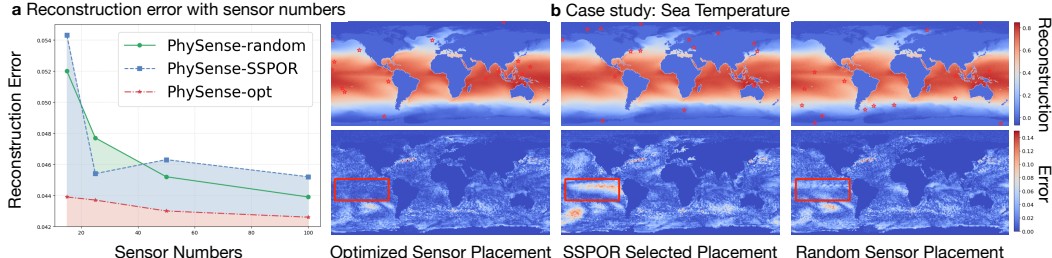

Figure 5: Different placement strategies comparison and case study on the sea temperature benchmark.

period. This also explains its poor performance on the turbulent flow, highlighting the limitations of traditional methods in capturing complex, non-stationary physical fields.

The optimized sensor placement improves performance by an additional 12.0% over the unoptimized counterpart in the low coverage setting (25 and 15 sensors). As shown in Fig. 5(a), the placement selected by SSPOR performs comparably to random selection, underscoring the importance of jointly co-optimizing reconstruction modeling and sensor placement optimization. Remarkably, just *15 optimally placed sensors*—covering *only 0.023% of the spatial domain*—are sufficient to achieve reconstruction accuracy comparable with the best settings on this benchmark, which highlights the feasibility of global ocean temperature field reconstruction with limited sensing resources.

Moreover, our placement strategy successfully avoids land regions and performs effective optimization under intricate land-sea boundaries (Fig. 5(b)). Such counterintuitive optimized placement is difficult to obtain through human-designed heuristics. The error map within the red box clearly indicates optimized placement captures more information, which in turn improves the reconstruction accuracy.

### 4.3 Car Aerodynamics

**Setups** This benchmark is the first to address physics sensing tasks on 3D irregular meshes. We select a fine 3D car model with 95,428 mesh points and simulate its pressure field using OpenFOAM (see Appendix C.1 for simulation details). The simulations span driving velocities from 20 to 40 m/s and yaw angles from -10 to 10 degrees. We simulate 100 cases in total, splitting them into 75 for training and 25 for testing. The task involves

Table 3: Main results on the car aerodynamics benchmark. Due to resource constraints, only the best deterministic baseline and generative baseline are selected. To support irregular meshes, the backbone of DiffusionPDE [18] is replaced with Transolver [48].

| Models | Car Aerodynamics | | | | |
|---|---|---|---|---|---|
| | #200 | #100 | #50 | # 30 | #15 |
| Senseiver [41] | 0.1009 | 0.1012 | 0.1018 | 0.1022 | 0.1053 |
| DiffusionPDE [18] | 0.0967 | 0.0966 | 0.0987 | 0.1055 | 0.2095 |
| **PhySense** | **0.0375** | **0.0382** | **0.0395** | **0.0416** | **0.0465** |
| **PhySense-opt** | **0.0369** | **0.0370** | **0.0370** | **0.0372** | **0.0386** |
| Promotion | 61.84% | 61.70% | 62.51% | 63.60% | 63.34% |

reconstructing the surface pressure field from 10 to 200 randomly sampled sensors placed exclusively on the vehicle surface, since placing sensors in the surrounding but off-surface regions presents practical challenges. This setup imposes particularly challenging conditions for placement optimization due to the complexity of the 3D surface and its geometric constraints.

**Results** As shown in Table 3, PhySense-opt consistently achieves state-of-the-art performance across all sensor counts (average improvement 62.6%). Moreover, our placement strategy proves effective, especially in low-coverage scenarios. With only 15 optimized sensors, we achieve a 17.0% improvement (from 0.0465 to 0.0386), and the reconstruction accuracy matches that of using 100 randomly placed sensors. Moreover, Fig. 1 illustrates that, since the wind approaches from the front of the driving vehicle, regions such as the front end and side mirrors experience significant pressure gradients, likely caused by flow separation or viscous effects. Our optimized sensor placement successfully captures these high-variation regions, validating the effectiveness of the projected gradient descent optimization. Furthermore, our optimized sensor placement can be deployed in real-world settings, such as wind tunnel experiments, to collect actual pressure data. This enables correction of discrepancies in simulation results and even of underlying physical systems, reducing the requirements for repeating costly real experiments and narrowing the sim-to-real gap.

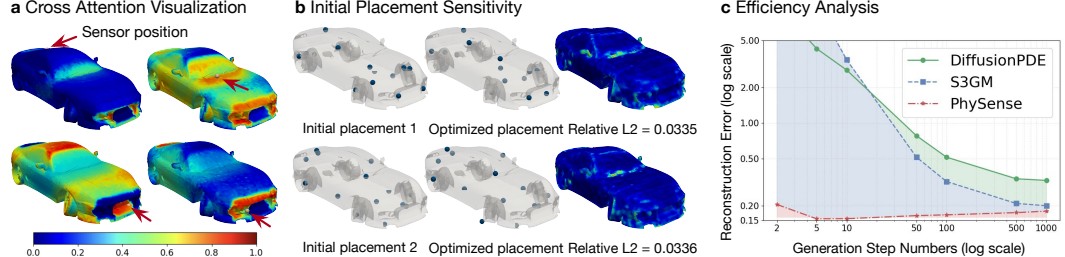

Figure 6: (a) The cross-attention map visualizations from the last layer of PhySense, where sensor positions are pointed by *red arrows*. (b) Initial placement sensitivity analysis shows that varying initial sensor placements converge to different but equally effective placements, achieving similar reconstruction accuracy. (c) Efficiency comparison on turbulent flow: these diffusion-based methods require 1,000 generation steps (**100× slower** than PhySense), but achieve even worse accuracy.

## 4.4 Model Analysis

**Ablations**  Beyond the main results, we explore how to best incorporate sparse sensor information. We compare our incorporation strategy with an alternative that concatenates an interpolated field—similar to the approaches used in VoroniCNN [14]. As shown in Table 4, this replacement leads to a significant performance drop, especially under low spatial coverage (e.g., 30 sensors), highlighting the advantage of our method in capturing critical information from scattered sensor observations.

Table 4: Ablation on sparse information incorporation via an interpolated field (*Rep. Inter.*).

| Ablations | Turbulent Flow | | | |
|---|---|---|---|---|
| | #200 | #100 | #50 | #30 |
| *Rep. Inter.* | 0.1571 | 0.2006 | 0.3584 | 0.7509 |
| Ours | **0.1233** | **0.1527** | **0.2586** | **0.5176** |

**Cross attention visualization**  To demonstrate how PhySense dynamically associates sensor observations with their relevant spatial influence regions, we visualize the cross-attention maps from the last cross attention block. As shown in Fig. 6(a), PhySense captures physically meaningful regions of influence for each sensor on the complex mesh, indicating the underlying physical structure for its high-fidelity reconstruction. Moreover, a highlighted pair in the second row shows two spatially adjacent sensors exhibit *diametrically opposed attention patterns* due to local geometric variations.

**Efficiency analysis**  The efficiency comparison on turbulent flow benchmark in Fig. 6(c) shows that our method achieves significantly lower reconstruction errors across all step counts. Notably, while DiffusionPDE and S³GM require over 50 steps to achieve the error below 0.5, our method consistently maintains a relative L2 error below 0.2 *even with as few as 2 steps*. This efficiency advantage stems from fundamental differences in sparse information incorporation strategies. DiffusionPDE and S³GM *indirectly* modify the data generation gradient based on sparse observations which require numerous iterative refinements to achieve proper guidance, while our approach *directly* conditions the model adaptively on sparse inputs during training. This architectural difference enables immediate and effective utilization of observational information from the very first inference step. As a result, we achieve better performance with *only 1% model evaluations* required by these baseline methods.

**Initial placement sensitivity**  During the sensor placement stage, different placement initializations lead to varying final sensor distributions. Meanwhile, as shown in Fig. 6(b), after placement optimization, the reconstruction performance remains *consistently high-quality* and comparable across these variants, indicating that our placement optimization process is not sensitive to different initializations.

**Progressive sensor placement optimization**  A key advantage of our flow-based reconstructor is its inherent support for progressive sensor placement optimization. Unlike non-generative models, which operate on static inputs (e.g., pure noise or zeros), our model processes a noise-data mixture $\mathbf{X}_t$ throughout the flow. This design inherently embeds a spectrum of reconstruction difficulties, corresponding to different noise levels, thereby amortizing the optimization process across the entire flow. Consequently, it provides more informative, tractable feedback for

Table 5: Ablation on progressive sensor placement optimization via a pure noise input.

| Ablations | Turbulent Flow | | | |
|---|---|---|---|---|
| | #200 | #100 | #50 | #30 |
| *noise-inut* | 0.1118 | 0.1293 | 0.1630 | 0.2286 |
| *mix-input* (origin) | **0.1106** | **0.1257** | **0.1558** | **0.2157** |
| Promotion | 1.07% | 2.78% | 4.42% | 5.65% |

Table 6: Performance comparison with four additional placement baselines on the turbulent flow benchmark, measured by relative L2 error. Results obtained with limited sensors (e.g., 30) are highlighted in blue, and the top-performing methods with comparable results are shown in **bold**.

| Placement Strategies | Senseiver | | | | PhySense | | | |
|---|---|---|---|---|---|---|---|---|
| | #200 | #100 | #50 | #30 | #200 | #100 | #50 | #30 |
| Random Sample | 0.1842 | 0.2316 | 0.3740 | 0.5746 | 0.1233 | 0.1527 | 0.2586 | 0.5176 |
| Grid Sample | 0.1615 | 0.2273 | 0.3845 | 0.5869 | 0.1235 | 0.1521 | 0.2436 | 0.4930 |
| Min-Max Sample | 0.1549 | 0.2091 | 0.3795 | 0.5747 | 0.1196 | 0.1533 | 0.3157 | 0.5022 |
| Our PGD on reconstruction model | **0.1388** | **0.1750** | **0.3025** | **0.4570** | **0.1106** | **0.1257** | **0.1558** | **0.2157** |
| Optimized Sensors from PhySense | **0.1533** | **0.1833** | **0.3035** | **0.4601** | **0.1106** | **0.1257** | **0.1558** | **0.2157** |

sensor placement. An ablation study shown in Table 5 confirms that replacing the noise-data *mix input* with pure *noise input* in the feedback loop leads to notable performance degradation, which is more pronounced with fewer sensors, underscoring the unique benefit of this flow-process feedback.

**Placement generalization** We compare our placement strategy with four more placement baselines across Senseiver and PhySense under different sensor numbers on turbulent flow dataset.

*(i) Grid*: We first place sensors on a uniform grid using the largest square number within the budget, and then randomly allocate the remainder. This strategy guarantees spatial coverage but remains agnostic to the physical dynamics. Practically, its performance is similar to random placement, aligning with classical sampling theory [4] that questions the efficacy of grid sampling for complex systems.

*(ii) Min-Max*: A heuristic method that promotes spread-out sensor placements by iteratively placing the next sensor at the point that maximizes the minimum distance to existing sensors. However, this approach does not yield significant improvements over random placement in our experiments.

*(iii) Our Projected Gradient Descent*: An upper-bound method using any reconstruction model (Senseiver or PhySense) and performs our projected gradient descent to optimize sensor placements.

*(iv) Optimized Sensor from PhySense*: To evaluate the *generality* of the learned placements by PhySense, we feed the placement into the Senseiver. The performance closely matches that of *Our PGD* and significantly outperforms all other baselines, particularly when the number of sensors is limited (e.g., 30), a challenging scenario to place sensors. This suggests that our learned placements effectively capture physically informative structures that *generalize beyond specific reconstructors*.

**Optimized sensor distribution visualization** To explore the distribution of *optimized sensors*, we visualize their relationship with *information distribution* (measured by the model's spatial gradient magnitude) and *variance distribution* (point-wise variance of the physical field) on turbulent flow benchmark. Interestingly, most of the optimized sensors lie within high-variance regions. More importantly, each sensor is located near a high-information point, and their spatial distributions exhibit similar patterns. This suggests that the optimizer is not merely clustering in high-variance areas but is strategically identifying spatially separated, high-sensitivity locations which capture the field's global dynamics. This aligns with the sensing principle that maximal information gain requires sampling diverse, non-redundant points rather than densely covering localized variance hotspots.

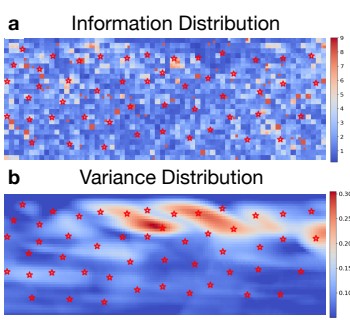

Figure 7: Visualization of information and variance distribution on turbulent flow benchmark.

## 5 Conclusion

To co-optimize physics field reconstruction and sensor placement for accurate physics sensing, this paper presents PhySense as a synergistic two-stage framework. The first stage develops a base model through flow matching that reconstructs dense physical fields from all feasible sensor placements. The second stage strategically optimizes sensor positions through projected gradient descent to respect geometry constraints. Furthermore, our theoretical analysis reveals a deep connection between the framework's objectives and the classical A-optimal principle, i.e., variance minimization. Extensive experiments demonstrate state-of-the-art performance across benchmarks, particularly on a complex 3D irregular dataset, while discovering informative sensor placements previously unconsidered.

## Acknowledgments and Disclosure of Funding

This work was supported by the National Natural Science Foundation of China (62021002 and U2342217), the BNRist Innovation Fund (BNR2024RC01010), State Grid Ningxia Electric Power Co. Science and Technology Project (SGNXYX00SCJS2400058), and the National Engineering Research Center for Big Data Software.

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

# A Proofs of Theorems in the Method Section

## A.1 Proof of Theorem 3.1

*Proof.* By the pointwise optimality of the squared loss in the training flow loss (1),

$$\mathbf{v}^*(\mathbf{x}, t, \mathbf{p}, \mathbf{X}_1(\mathbf{p})) \; = \; \mathbb{E}\big[\mathbf{X}_1 - \mathbf{X}_0 \; \big| \; \mathbf{X}_t = \mathbf{x}, t, \mathbf{p}, \mathbf{X}_1(\mathbf{p})\big].$$

Taking the conditional expectation of $\bar{\mathbf{X}}_1$ gives

$$\mathbb{E}\big[\bar{\mathbf{X}}_1 \mid \mathbf{p}, \mathbf{X}_1(\mathbf{p})\big] = \mathbb{E}\Big[\mathbf{X}_0 + \int_0^1 \mathbf{v}^*(\mathbf{X}_t, t, \mathbf{p}, \mathbf{X}_1(\mathbf{p})) \, \mathrm{d}t \; \Big| \; \mathbf{p}, \mathbf{X}_1(\mathbf{p})\Big]$$

$$= \underbrace{\mathbb{E}[\mathbf{X}_0 \mid \mathbf{p}, \mathbf{X}_1(\mathbf{p})]}_{(A)} + \underbrace{\mathbb{E}\Big[\int_0^1 \mathbf{v}^*(\mathbf{X}_t, t, \mathbf{p}, \mathbf{X}_1(\mathbf{p})) \, \mathrm{d}t \; \Big| \; \mathbf{p}, \mathbf{X}_1(\mathbf{p})\Big]}_{(B)}.$$

**Step 1:** Move the expectation inside the integral (Fubini's theorem):

$$(B) \; = \; \int_0^1 \mathbb{E}\Big[\mathbf{v}^*(\mathbf{X}_t, t, \mathbf{p}, \mathbf{X}_1(\mathbf{p})) \; \Big| \; \mathbf{p}, \mathbf{X}_1(\mathbf{p})\Big] \, \mathrm{d}t.$$

**Step 2:** Substitute the definition of $\mathbf{v}^*$ and apply the law of total expectation:

$$\mathbb{E}\Big[\mathbf{v}^*(\mathbf{X}_t, t, \mathbf{p}, \mathbf{X}_1(\mathbf{p})) \; \Big| \; \mathbf{p}, \mathbf{X}_1(\mathbf{p})\Big] = \mathbb{E}\Big[\mathbb{E}\big[\mathbf{X}_1 - \mathbf{X}_0 \mid \mathbf{X}_t, t, \mathbf{p}, \mathbf{X}_1(\mathbf{p})\big] \; \Big| \; \mathbf{p}, \mathbf{X}_1(\mathbf{p})\Big]$$

$$= \mathbb{E}\big[\mathbf{X}_1 - \mathbf{X}_0 \mid \mathbf{p}, \mathbf{X}_1(\mathbf{p})\big].$$

The right-hand side no longer depends on $t$, hence

$$(B) \; = \; \int_0^1 \mathbb{E}\big[\mathbf{X}_1 - \mathbf{X}_0 \mid \mathbf{p}, \mathbf{X}_1(\mathbf{p})\big] \mathrm{d}t \; = \; \mathbb{E}\big[\mathbf{X}_1 - \mathbf{X}_0 \mid \mathbf{p}, \mathbf{X}_1(\mathbf{p})\big].$$

**Step 3:** Combine $(A)$ and $(B)$:

$$(A) + (B) \; = \; \mathbb{E}\big[\mathbf{X}_0 \mid \mathbf{p}, \mathbf{X}_1(\mathbf{p})\big] + \mathbb{E}\big[\mathbf{X}_1 - \mathbf{X}_0 \mid \mathbf{p}, \mathbf{X}_1(\mathbf{p})\big] \; = \; \mathbb{E}\big[\mathbf{X}_1 \mid \mathbf{p}, \mathbf{X}_1(\mathbf{p})\big],$$

Therefore, flow-based reconstructor is an unbiased estimator of physial fields. □

## A.2 Proof of Theorem 3.5

**Part 1.** First, we will prove $\mathcal{L}_A(\mathbf{p}) = \mathbb{E}\big[\|\mathbf{X}_1 - \mathbb{E}\mathbf{X}_1\|^2\big] = \sum_{i=1}^d \lambda_i(\mathbf{p})$ with eigenvalues $\{\lambda_i(p)\}_{i=1}^d$ of $\Sigma_p$.

Because $\mathbf{X}_1 \mid (\mathbf{p}, \mathbf{X}_1(\mathbf{p})) \sim \mathcal{N}(\mathbf{0}, \Sigma_{\mathbf{p}})$, we have

$$\mathcal{L}_A(\mathbf{p}) \; = \; \mathbb{E}\big[\|\mathbf{X}_1 - \mathbb{E}\mathbf{X}_1\|^2\big] \; = \; \mathrm{tr}\big(\mathrm{Var}(\mathbf{X}_1)\big) \; = \; \mathrm{tr}\big(\Sigma_{\mathbf{p}}\big).$$

Let the *orthogonal decomposition* of the placement–dependent covariance matrix be

$$\Sigma_{\mathbf{p}} = U\Lambda U^{\mathsf{T}}, \quad \Lambda = \mathrm{diag}\big(\lambda_1(\mathbf{p}), \ldots, \lambda_d(\mathbf{p})\big),$$

where $U$ is an orthogonal matrix, i.e. $UU^{\mathsf{T}} = I$. where $U$ is an orthogonal matrix, i.e. $UU^{\mathsf{T}} = I$. All eigenvalues $\lambda_i(p)$ are non-negative because the covariance matrix $\Sigma_p$ is positive semidefinite. Since the trace is invariant under orthogonal similarity transformations, we obtain

$$\mathcal{L}_A(\mathbf{p}) \; = \; \mathrm{tr}\big(\Sigma_{\mathbf{p}}\big) \; = \; \mathrm{tr}(\Lambda) \; = \; \sum_{i=1}^d \lambda_i(\mathbf{p}). \tag{A.0}$$

**Part 2.** Second, we will show,

$$\mathcal{L}_{\text{flow}}(\mathbf{p}) = \int_0^1 \mathbb{E}\Big[\big\|(\mathbf{X}_1 - \mathbf{X}_0) - \mathbf{v}^*(\mathbf{X}_t, t, \mathbf{p}, \mathbf{X}_1(\mathbf{p}))\big\|^2\Big]\, dt$$

$$\stackrel{\text{def}}{=} \int_0^1 \text{tr}\Big(\text{Var}\big(\mathbf{X}_1 - \mathbf{X}_0 \mid \mathbf{X}_t, t, \mathbf{p}, \mathbf{X}_1(\mathbf{p})\big)\Big)$$

$$= \sum_{i=1}^d \frac{\pi}{2}\sqrt{\lambda_i(\mathbf{p})},$$

**Step 1:** Calculate the conditional variance $\text{Var}\big(\mathbf{X}_1 - \mathbf{X}_0 \mid \mathbf{X}_t, t, \mathbf{p}, \mathbf{X}_1(\mathbf{p})\big)$

We denote that,

$$\mathbf{Z} = \begin{pmatrix} \mathbf{X}_0 \\ \mathbf{X}_1 \end{pmatrix}, \quad \Sigma_{\mathbf{Z}} = \text{diag}(I, \Sigma_{\mathbf{p}}), \quad \mathbf{Y} = \mathbf{X}_1 - \mathbf{X}_0, \quad \mathbf{X}_t = (1-t)\mathbf{X}_0 + t\mathbf{X}_1 \ (0 \leq t \leq 1).$$

Since $\mathbf{X}_0$ and $\mathbf{X}_1$ are independent, $\mathbf{Z}$ conditioned on $(\mathbf{p}, \mathbf{X}_1(\mathbf{p}))$ remains Gaussian,

$$\mathbf{Z} \mid (\mathbf{p}, \mathbf{X}_1(\mathbf{p})) \sim \mathcal{N}(\mathbf{0}, \Sigma_{\mathbf{Z}}).$$

Writing $\mathbf{Y}$ and $\mathbf{X}_t$ as *linear* maps of $\mathbf{Z}$,

$$\mathbf{Y} = V\mathbf{Z}, \qquad V = \big(-I, \ I\big), \qquad \mathbf{X}_t = C\mathbf{Z}, \qquad C = \big((1-t)I, \ tI\big),$$

thus the pair $(\mathbf{Y}, \mathbf{X}_t)$ is jointly Gaussian. The standard conditional–covariance formula yields

$$\text{Var}\big(\mathbf{Y} \mid \mathbf{X}_t, t, \mathbf{p}, \mathbf{X}_1(\mathbf{p})\big) = V\Sigma_{\mathbf{Z}}V^{\mathsf{T}} - V\Sigma_{\mathbf{Z}}C^{\mathsf{T}}\big(C\Sigma_{\mathbf{Z}}C^{\mathsf{T}}\big)^{-1}C\Sigma_{\mathbf{Z}}V^{\mathsf{T}} \qquad \text{(A.1)}$$

with

$$V\Sigma_{\mathbf{Z}}V^{\mathsf{T}} = I + \Sigma_{\mathbf{p}},$$

$$C\Sigma_{\mathbf{Z}}C^{\mathsf{T}} = (1-t)^2 I + t^2 \Sigma_{\mathbf{p}},$$

$$V\Sigma_{\mathbf{Z}}C^{\mathsf{T}} = -(1-t)I + t\,\Sigma_{\mathbf{p}}.$$

Substituting into (A.1) gives

$$\text{Var}\big(\mathbf{Y} \mid \mathbf{X}_t, t, \mathbf{p}, \mathbf{X}_1(\mathbf{p})\big) = I + \Sigma_{\mathbf{p}} - \big(t\Sigma_{\mathbf{p}} - (1-t)I\big)\big((1-t)^2 I + t^2 \Sigma_{\mathbf{p}}\big)^{-1}\big(t\Sigma_{\mathbf{p}} - (1-t)I\big) \quad \text{(A.2)}$$

**Step 2:** Diagonalisation and scalar reduction

Recall from (A.2) the matrix form of the conditional variance $\text{Var}\big(\mathbf{Y} \mid \mathbf{X}_t, t, \mathbf{p}, \mathbf{X}_1(\mathbf{p})\big)$ and denote it by $\Gamma_t(\Sigma_{\mathbf{p}})$. Because $\Sigma_{\mathbf{p}} = U\Lambda U^{\mathsf{T}}$ for a orthogonal matrix $U$ and a diagonal matrix $\Lambda = \text{diag}(\lambda_1, \ldots, \lambda_d)$, **all terms in** (A.2) **commute with** $U$; consequently

$$\Gamma_t(\Sigma_{\mathbf{p}}) = U\,\Gamma_t(\Lambda)\,U^{\mathsf{T}}.$$

Thus,

$$\text{tr}(\Gamma_t(\Sigma_{\mathbf{p}})) = \text{tr}(\Gamma_t(\Lambda)).$$

Consequently, the flow–matching objective can be written entirely in terms of the diagonal matrix $\Lambda$:

$$\mathcal{L}_{\text{flow}}(\mathbf{p}) = \int_0^1 \text{tr}\Big(\text{Var}\big(\mathbf{X}_1 - \mathbf{X}_0 \mid \mathbf{X}_t, t, \mathbf{p}, \mathbf{X}_1(\mathbf{p})\big)\Big)\, dt$$

$$= \int_0^1 \text{tr}\big(\Gamma_t(\Sigma_{\mathbf{p}})\big)\, dt$$

$$= \int_0^1 \text{tr}\big(\Gamma_t(\Lambda)\big)\, dt$$

$$= \sum_{i=1}^d \int_0^1 \Gamma_t(\lambda_i(\mathbf{p}))\, dt,$$

where the last equality uses that $\Gamma_t(\Lambda) = \mathrm{diag}(\Gamma_t(\lambda_1), \ldots, \Gamma_t(\lambda_d))$ is diagonal, so its trace equals the sum of its diagonal entries.

According to the expression of $\Gamma_t$,

$$\mathcal{L}_{\text{flow}}(\mathbf{p}) = \sum_{i=1}^{d} \int_0^1 \frac{\lambda_i}{t^2\lambda_i + (1-t)^2} \, dt, \quad \lambda_i \geq 0 \tag{A.3}$$

**Step 3:** Calculate the integral above over $t \sim \mathcal{U}[0,1]$.

For $\lambda > 0$ and consider

$$I(\lambda) := \int_0^1 \frac{\lambda}{t^2\lambda + (1-t)^2} \, dt.$$

Set $A := \lambda + 1$. Completing the square in the denominator gives

$$\lambda t^2 + (1-t)^2 = (\lambda+1)t^2 - 2t + 1 = A\left(t - \tfrac{1}{A}\right)^2 + \frac{\lambda}{A}.$$

**First substitution.** We introduce

$$u = \sqrt{\frac{A}{\lambda}}\left(t - \tfrac{1}{A}\right), \qquad dt = \sqrt{\frac{\lambda}{A}} \, du,$$

so that

$$t = 0 \implies u_0 = -\frac{1}{\sqrt{\lambda A}}, \qquad t = 1 \implies u_1 = \frac{\sqrt{\lambda}}{\sqrt{A}}.$$

Substituting, we obtain

$$I(\lambda) = \sqrt{\frac{\lambda}{A}} \int_{u_0}^{u_1} \frac{du}{u^2 + A^{-1}} = \sqrt{\lambda A} \int_{u_0}^{u_1} \frac{du}{Au^2 + 1}.$$

**Second substitution.** Absorb the constant $A^{-1}$ by setting

$$v = \sqrt{A}u, \qquad du = \frac{dv}{\sqrt{A}},$$

which converts the integrand to the standard form $(v^2 + 1)^{-1}$. The limits become

$$v_0 = \sqrt{A}\,u_0 = -\frac{1}{\sqrt{\lambda}}, \qquad v_1 = \sqrt{A}\,u_1 = \sqrt{\lambda}.$$

Collecting the Jacobian factors yields

$$I(\lambda) = \sqrt{\lambda A}\frac{1}{\sqrt{A}} \int_{v_0}^{v_1} \frac{dv}{v^2 + 1} = \sqrt{\lambda} \int_{-1/\sqrt{\lambda}}^{\sqrt{\lambda}} \frac{dv}{v^2 + 1},$$

The integrand is elementary, giving

$$\begin{aligned}
I(\lambda) &= \sqrt{\lambda} \left[\arctan u\right]_{u=-1/\sqrt{\lambda}}^{u=\sqrt{\lambda}} \\
&= \sqrt{\lambda}\left(\arctan\sqrt{\lambda} - \arctan-\tfrac{1}{\sqrt{\lambda}}\right) \\
&= \sqrt{\lambda}\left(\arctan\sqrt{\lambda} + \arctan\tfrac{1}{\sqrt{\lambda}}\right).
\end{aligned}$$

For any $x > 0$ the identity $\arctan x + \arctan\frac{1}{x} = \pi/2$ holds. Then substituting $x = \sqrt{\lambda}$ yields,

$$I(\lambda) = \frac{\pi}{2}\sqrt{\lambda}.$$

For $\lambda = 0$, the identity also holds obviously.

Applying this to every $\lambda_i$ in (A.3) gives

$$\mathcal{L}_{\text{flow}}(\mathbf{p}) = \sum_{i=1}^{d} \frac{\pi}{2} \sqrt{\lambda_i(\mathbf{p})} \,,$$

which is the target of Part 2.

**Part 3.** We prove $\frac{2}{\pi^2} \mathcal{L}_{\text{flow}}^2(p) \leq \mathcal{L}_A(p) \leq \frac{4}{\pi^2} \mathcal{L}_{\text{flow}}^2(p)$.

By the arithmetic–geometric mean inequality, we have,

$$\sum_{i=1}^{d} \lambda_i \leq \left( \sum_{i=1}^{d} \sqrt{\lambda_i(\mathbf{p})} \right)^2 \leq 2 \sum_{i=1}^{d} \lambda_i \quad \Longrightarrow \quad \frac{1}{2} \left( \sum_{i=1}^{d} \sqrt{\lambda_i(\mathbf{p})} \right)^2 \leq \mathcal{L}_A \leq \left( \sum_{i=1}^{d} \sqrt{\lambda_i(\mathbf{p})} \right)^2 .$$

Therefore,

$$\frac{2}{\pi^2} \mathcal{L}_{\text{flow}}^2(\mathbf{p}) \leq \mathcal{L}_A(\mathbf{p}) \leq \frac{4}{\pi^2} \mathcal{L}_{\text{flow}}^2(\mathbf{p}) \,,$$

which completes **Part 3** and hence the proof of Theorem 3.5. □

**Remark A.1.** *Furthermore, under Assumption 3.2, this mutual control relationship naturally extends to both D-optimality and E-optimality objectives. Specifically, their formulations simplify to*

$$\mathcal{L}_D(\mathbf{p}) := \det(\Sigma_{\mathbf{p}}) = \prod_{i=1}^{d} \lambda_i(\mathbf{p}), \ \mathcal{L}_E(\mathbf{p}) := \lambda_{\max}(\Sigma_{\mathbf{p}}). \tag{3}$$

*It is evident that both $\mathcal{L}_D$ and $\mathcal{L}_E$ can be polynomially bounded above and below by $\mathcal{L}_A$ due to standard spectral inequalities. Consequently, through the control of $\mathcal{L}_A$, the flow-loss objective also polynomially controls $\mathcal{L}_D$ and $\mathcal{L}_E$ in both directions. Therefore, the flow-loss objective serves as a spectrum-aware surrogate that can also guide D-optimal and E-optimal sensor placement with bounded distortion.*

## B  Full Results of Sensor Placement Optimization

Due to the space limitation of the main text, we present the full results of sensor placement results on turbulent flow and sea temperature here, as a supplement to Fig. 4 and Fig. 5.

Given the same high-capacity reconstruction model, our optimized placement consistently achieves superior performance compared to SSPOR and random baselines, confirming the effectiveness of our joint strategy. Interestingly, SSPOR does not always outperform random sampling; for example, on the sea temperature dataset, it often performs slightly worse. This underscores the importance of co-optimizing sensor placement alongside the reconstruction process.

Table 7: Performance under different placement strategies on turbulent flow and sea temperature.

| Models | Placement Strategy | Turbulent Flow | | | | Sea Temperature | | | |
|---|---|---|---|---|---|---|---|---|---|
| | | #200 | #100 | #50 | #30 | #100 | #50 | #25 | #15 |
| PhySense | Random | 0.1233 | 0.1527 | 0.2586 | 0.5176 | 0.0439 | 0.0452 | 0.0477 | 0.0520 |
| PhySense | SSPOR | 0.1143 | 0.1348 | 0.1815 | 0.3397 | 0.0452 | 0.0463 | 0.0454 | 0.0543 |
| PhySense | **Optimized** | **0.1106** | **0.1257** | **0.1558** | **0.2157** | **0.0426** | **0.0430** | **0.0437** | **0.0439** |
| Relative Promotion | | 3.23% | 6.75% | 14.16% | 36.52% | 2.96% | 4.87% | 3.74% | 15.58% |

## C  Implementation Details

In this section, we provide the implementation details of our experiments, including **benchmarks, metrics, and implementations**. All the experiments are conducted based on PyTorch 2.1.0 [34] and on a A100 GPU server with 144 CPU cores.

## C.1 Benchmarks

While the turbulent flow and sea temperature benchmarks are described in detail in the main text, here we focus on elaborating the key details of *our generated* car aerodynamic benchmark.

**a** Computation domain

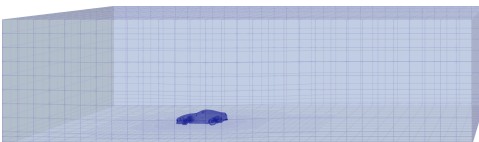

**b** Slice of the refined mesh

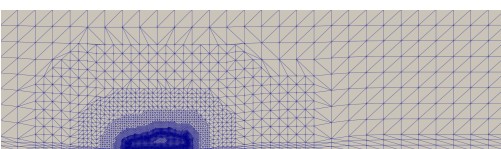

Figure 8: (a) Computational Domain of the CFD simulation. (b) Refinement regions around the surface.

We use a watertight 3D car mesh taken from the ShapeNet V1 [6] dataset, which contains around 50k triangular facets. The aerodynamic pressure of the car surface is simulated using OpenFOAM [20]. As shown in Fig. 8(a), we create a virtual wind tunnel to simulate the airflow around a car. The width, length and height of the wind tunnel are set to 16 meters, 30 meters and 8 meters respectively. The computational mesh around the car surface is refined to a smaller mesh size for more accurate aerodynamics modeling, which is illustrated in Fig. 8(b).

We specify different inlet velocities and yaw angles for different simulation cases. The velocity ranges from 20 $m/s$ to 40 $m/s$, while the yaw angles range from -10 degrees to 10 degrees. We utilize Reynolds-Averaged Simulation (RAS) with the k-omega SST turbulence model in OpenFOAM. 600 simulation iterations are conducted for each case to ensure convergence. A no-slip boundary condition of the velocity field is defined on the car surface. We simulate 100 cases in total, splitting them into 75 for training and 25 for testing.

## C.2 Metrics

To evaluate the accuracy of the reconstructed physical field, we employ the widely used *relative L2* metric, defined as

$$\text{Relative } L_2 = \frac{\|\mathbf{u} - \hat{\mathbf{u}}\|_2}{\|\mathbf{u}\|_2},$$

where $\mathbf{u}$ denotes the reference (ground-truth) field and $\hat{\mathbf{u}}$ represents the model reconstruction.

## C.3 Implementations

We consider the following baselines on the turbulent flow and sea temperature benchmarks:

- **Reconstruction models:** SSPOR, VoronoiCNN, Senseiver, S³GM, and DiffusionPDE.
- **Sensor placement strategies:** random sampling and the SSPOR's sensor selection method.

Due to the geometric complexity of the car's 3D surface, we compare it against the top two performing reconstruction baselines—Senseiver and DiffusionPDE—for this task. The common setups for all models are shown in the Table 8.

Table 8: Common setups for all models in three benchmarks. Other setups follow their original setup.

| Benchmarks | Turbulent Flow | Sea Temperature | Car Aerodynamics |
|---|---|---|---|
| Epochs | 800 | 300 | 300 |
| Batch Size | 60 | 40 | 1 |
| Learning Rate | $10^{-3}$ | $2 \times 10^{-4}$ | $10^{-3}$ |
| Optimizer | | ADAM | |

**SSPOR [32]** We utilize the official implementation of SSPOR, the PySensor package [8] to evaluate the reconstruction and sensor selection performance of SSPOR. SSPOR employs POD [2] to express

high-dimensional data as a linear combination of several orthonormal eigenmodes obtained from the singular value decomposition (SVD). Specifically, we employ the `pysensors.SSPOR` class with an SVD basis to obtain the targeted number of sensors from the training data. The selected sensors are then used to reconstruct dense fields from sparse observations either with SSPOR or our method.

**VoronoiCNN [14]** VoronoiCNN integrates Voronoi tessellation with convolutional neural networks (CNNs). However, the official implementation uses a simplistic CNN that struggles with complex and dynamic fields. We replace it with a more powerful U-Net [38] architecture, using 64 base feature channels, with 4 downsampling and upsampling layers and skip connections to enhance information flow. Due to its inability to handle irregular car aerodynamic meshes, experiments are conducted only on turbulent flow and sea temperature datasets under a unified setup.

**Senseiver [41]** We utilize the official implementation of Senseiver. Senseiver is built upon an implicit neural representation (INR) architecture, which optimizes reconstructions at individual spatial locations rather than over the entire physical field. Specifically, during training, 2048 points are randomly sampled from each field to supervise the model. This inherent sampling randomness further impacts the stability of Senseiver and contributes to its slower convergence. To better assess its full potential, we did not strictly enforce the same training epoch limit used for other baselines. Instead, we allowed Senseiver to train until the default early stopping criterion in its official implementation was triggered. Across the three distinct benchmarks we use, the hyperparameters of Senseiver architecture are carefully tuned based on the official implementation scripts.

**S$^3$GM [28]** In our experiments, we follow the official implementation and conduct training and inference. Its backbone is the full U-Net architecture equipped with attention mechanisms and timestep embeddings. We find that when the number of iterative optimization steps is less than 100, S$^3$GM fails to produce clear generation results. Moreover, the generation results can be significantly affected by the random sampling strategy of the sensors, with some cases exhibiting abnormally high relative errors, even greater than 10. Therefore, we increase the number of iterative optimization steps to 1,000 to obtain comparable experimental results.

**DiffusionPDE [18]** We borrow the model architectures of the original implementation of DiffusionPDE for the turbulent flow and sea temperature benchmarks, and employ a Transolver [48] as the backbone for the unstructured car aerodynamics benchmark. For inference, we also follow the original implementation configurations, using a total of 2,000 steps for better performance. The guidance weights we search in each benchmark are shown in Table 9.

Table 9: Guidance weight of the observation loss for DiffusionPDE. Our final selection is **bolded**

| Benchmarks | Turbulent Flow | Sea Temperature | Car Aerodynamics |
|---|---|---|---|
| Guidance Weight | {1000, **2500**, 5000} | {1000, **2500**, 5000} | {1000, 2500, **5000**} |

**PhySense** For the reconstruction model, we use DiT as base models for turbulent flow and sea temperature benchmarks, and switch to Transolver for car aerodynamics benchmark. The detailed model configurations are summarized in Table 10.

Table 10: PhySense model configurations for different benchmarks. "–" in the patch size column indicates that Transolver does not apply the patchify operation.

| Benchmark | Patch Size | dim | Depth | Heads | dim_head | mlp_dim | Others |
|---|---|---|---|---|---|---|---|
| Turbulent Flow | (2, 2) | 374 | 8 | 8 | 32 | 374 | – |
| Sea Temperature | (3, 3) | 374 | 12 | 8 | 32 | 374 | – |
| Car Aerodynamics | – | 374 | 12 | 8 | 32 | 374 | slice_num=32 |

For sensor placement optimization, we adopt the Adam optimizer to perform gradient-based updates, followed by a projection step implemented via nearest-neighbor search to satisfy the spatial constraints. All models are trained for 5 epochs with a cosine learning rate decay scheduler. The learning rate is *closely tied to the spatial scale of the physical domain* and is carefully tuned for each setting, as summarized in Table 11.

Table 11: Learning rate of sensor placement optimization on three benchmarks.

| Benchmarks | Turbulent Flow | Sea Temperature | Car Aerodynamics |
|---|---|---|---|
| Learning rate | 0.25 | 1 | 0.0025 |

# D  Additional Analysis

## D.1  Comparsion with AdaLN-Zero

Here, we compare our sparse sensor incorporation strategy on turbulent flow with a widely used conditioning approach in diffusion models, namely the AdaLN-Zero block in DiT [35]. As shown in Table 12, replacing our design with AdaLN-Zero leads to a substantial degradation in performance, with reconstruction errors approaching 1. This is because sensor placement inherently captures localized information, whereas AdaLN-Zero aggregates all sensor data into a single global variable, which is insufficient for preserving the spatial specificity required in reconstruction tasks.

Table 12: Ablation on sparse information incorporation via AdaLN-Zero block, i.e., *Rep. AdaLN-Zero*

| Sensor Number | #200 | #100 | #50 | #30 |
|---|---|---|---|---|
| *Rep. AdaLN-Zero* | 0.9179 | 0.9486 | 0.9695 | >1 |
| Ours | **0.1233** | **0.1527** | **0.2586** | **0.5176** |

## D.2  Hyperparameter Sensitivity

As we adopt projected gradient descent to optimize sensor placement, the learning rate $\eta$ in Eq. (2) becomes a critical factor, especially when operating under complex geometric and high-dimensional surface constraints. We investigate the learning rate sensitivity on the car aerodynamics benchmark. As shown in Table 13, the optimization process is relatively insensitive to learning rate variations when the number of sensors exceeds 30. In contrast, the 15-sensor setting—covering only 0.016% of the spatial domain—exhibits noticeable sensitivity, indicating the increased difficulty of optimization under extreme sparsity. We ultimately select $\eta = 0.0025$ as the final choice.

Table 13: Impact of learning rate on sensor placement optimization in the car aerodynamics benchmark. We ultimately select $\eta = 0.0025$ as the final choice, which is **bolded**.

| Learning rate | 0.005 | **0.0025** | 0.001 | 0.0005 |
|---|---|---|---|---|
| 30 sensors | 0.0379 | **0.0372** | 0.0369 | 0.0372 |
| 15 sensors | 0.0423 | **0.0386** | 0.0389 | 0.0400 |

## D.3  Statistical Analysis

Table 14: Evaluation results of the PhySense reconstruction model under *random sensor placements* with varying sensor counts (10–200) on car aerodynamics benchmark. For each setting, the reconstruction is repeated three times, and the mean and variance of the relative L2 loss are reported.

| Sensor Number | #200 | #100 | #50 | #30 | #15 |
|---|---|---|---|---|---|
| Sampling 1 | 0.0376 | 0.0386 | 0.0396 | 0.0414 | 0.0458 |
| Sampling 2 | 0.0379 | 0.0378 | 0.0387 | 0.0414 | 0.0436 |
| Sampling 3 | 0.0370 | 0.0382 | 0.0401 | 0.0419 | 0.0501 |
| Mean | **0.0375** | **0.0382** | **0.0395** | **0.0416** | **0.0465** |
| Standard derivation | 0.0004 | 0.0003 | 0.0006 | 0.0002 | 0.0027 |

In this section, we analyze the statistical significance of our experimental results. Our pipeline consists of two stages: training the reconstruction model and optimizing the sensor placement. Due to the high computational cost of training the base reconstruction model, we train it only once. The sensitivity of sensor placement initialization has been discussed in Section 4.4. Our findings show that while different initializations result in different placements, each optimized placement is effective.

Table 15: Evaluation results of the PhySense reconstruction model under *optimized sensor placements* with varying sensor counts (10–200) on car aerodynamics benchmark. For each setting, the reconstruction is repeated three times, and the mean and variance of the relative L2 loss are reported.

| Sensor Number | #200 | #100 | #50 | #30 | #15 |
|---|---|---|---|---|---|
| Sampling 1 | 0.0369 | 0.0371 | 0.0369 | 0.0370 | 0.0378 |
| Sampling 2 | 0.0367 | 0.0372 | 0.0369 | 0.0374 | 0.0389 |
| Sampling 3 | 0.0370 | 0.0368 | 0.0373 | 0.0373 | 0.0390 |
| Mean | **0.0369** | **0.0370** | **0.0370** | **0.0372** | **0.0386** |
| Standard derivation | 0.0001 | 0.0002 | 0.0002 | 0.0002 | 0.0005 |

Furthermore, since our task is a reconstruction problem, and our reconstruction model essentially learns a conditional distribution, we report in the table above the mean and standard deviation of reconstruction loss under multiple samplings on the car aerodynamics benchmark, across varying numbers of sensors. As shown in Table 14 and 15, when the number of sensors exceeds 15, the standard deviation becomes relatively small, indicating stable performance. Moreover, regardless of the sensor count, once the placement is optimized, the reconstruction results exhibit consistent stability across multiple runs.

## D.4 Additional Metric for Gradients

Table 16: Comparison on sea temperature benchmark. Relative L2 error of gradient fields is reported.

| Sea Temperature (relative L2 of gradient field) | #100 | #50 | #25 | #15 |
|---|---|---|---|---|
| Senseiver + random sensor placement | 0.5456 | 0.5470 | 0.5501 | 0.5515 |
| PhySense + random sensor placement | **0.4877** | **0.4884** | **0.4890** | **0.4892** |
| Relative promotion | 11.87% | 11.98% | 12.51% | 12.74% |

In this section, we additionally evaluate *the relative L2 error of the gradient fields* on the sea temperature to better reflect whether the model captures meaningful physical fidelity. As shown in the Table 16, PhySense achieves significantly lower gradient error compared to the second-best model, Senseiver, demonstrating its superior spatial physical fidelity.

## D.5 Uncertainty quantification

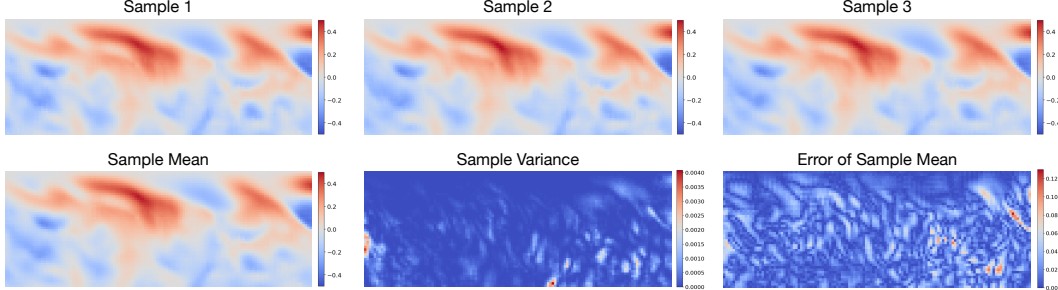

Figure 9: Visualization of uncertainty quantification for the flow-based reconstruction model on the turbulent flow benchmark. The sample variance is notably smaller than the physical field and error.

Flow-based reconstruction models explicitly learn the full conditional data distribution, which inherently facilitates uncertainty quantification. To complete this, we reconstruct each input three times, each with a distinct random noise initialization. Subsequently, we compute the pixel-wise sample mean and sample variance across these reconstructions. As illustrated in Fig. 9, the magnitude of the variance is *orders of magnitude smaller* than that of the sample mean and the reconstruction error. This indicates high confidence in our model, as it consistently produces similar reconstructions irrespective of the initial noise vector.

## E    Limitations and Future Work

While our current formulation assumes point-wise sensor observations, this may not fully capture the behavior of certain real-world sensors that integrate information over small spatial neighborhoods. For example, radar and precipitation sensors typically respond to local regions rather than single points. Incorporating regional observation into both the reconstruction model and the sensor placement optimization could further improve the effectiveness of our approach. As a future direction, we aim to extend both our model and theoretical framework from point-wise observations to region-based sensing, which better reflects the behavior of many real-world sensors.

## F    Boarder Impacts

We introduce PhySense, a synergistic two-stage framework for accurate physics sensing with theorectical guarantees. Extensive experiments validate its superior reconstruction accuracy and informative sensor placements. Beyond performance gains, our work offers a new perspective on the sensor placement problem from a deep learning standpoint. Thus, PhySense may inspire some future research in relevant domains. Moreover, our optimized placement could potentially inform how sensors are placed in more applications, enhancing their reconstruction accuracy.

Since our work is purely focused on algorithmic design for accurate physics sensing, there are no potential negative social impacts or ethical risks.

