# OpenReview forum: "PhySense: Sensor Placement Optimization for Accurate Physics Sensing"
_NeurIPS.cc/2025/Conference — NeurIPS 2025 oral_

### Official Review · Reviewer_ewdF · 2025-06-19

**Clarity:** 2
**Significance:** 3
**Originality:** 3
**Rating:** 4
**Confidence:** 2

**Summary:**

The paper presents a framework for efficient reconstruction of physical fields. This is done by training a flow-based generative model which can reconstruct the physical field from sparse observations, and a method to select the optimal sensor readings to select based on a differentiable objective function that can be optimised with projected GD.

**Questions:**

1. How efficient to train the generative models in the first place, both in terms of the time required and the number of training samples or simulations that have to be used?

2. How would the other sparse reconstruction methods work if a better sensor placement is used? This could potentially make the use of flow models more convincing.

3. Related, why would a flow based model be required in this case? Other alternatives apart from those discussed in the paper would be things such as some encoder/decoder model (that can also quite directly generate the reconstructed field), traditional numerical solvers or even PINNs or PINOs which are known to work for inverse problems. I suspect one of the reason may be the uncertainty quantification although this is not clearly demonstrated in the presented results. Other reasons should also exist which would make the paper motivation stronger.

4. Are the selected sensor placements interpretable or have some characteristics in any way? Furthermore, how sensitive are they to the GD process -- i.e., if I initialise my optimisation differently would the recovered sensor placements change?

**Ethical Concerns:**

["NO or VERY MINOR ethics concerns only"]

**Final Justification:**

See response to the author. Mainly I think the work is likely valuable for the context of generative models, but I am less confident about comparing this work against those since I am not familiar with other works in the area.

**Limitations:**

Some limitations are discussed.

**Paper Formatting Concerns:**

None.

**Quality:**

2

**Strengths And Weaknesses:**

Strengths:
- The proposed problem seems to have real use case and is of interesting scope. The proposed solution is also interesting.
- The benchmarks used seem to be of sufficient difficulty (although it would also be interesting to see more realistic data used instead of simulated ones).
- Theoretical results seem to be sufficient for the claims that are made by the paper. The link to OED methods is also a good point.

Weaknesses:
- The reported results have no error bars or confidence intervals (e.g., across the different reconstructed field cases), making it difficult to evaluate how statistically significant the results are.
- Since the paper does also seem to focus on the sensor placement aspect, it would be interesting to see how other point selection method would work in this problem (e.g., grid placement or some configuration where the sensors are more spread out, or selection based on other ED optimality criteria), and how those placements may affect the other reconstruction methods proposed. This aspect is quite crucial to provide a more convincing and fair comparison with other methods.
- The motivations could probably be discussed more explicitly, especially with why a quick reconstructor or even a flow-based model specifically needs to be used in this case (I don't doubt that there isn't one, but that the paper would be more complete if that was discussed).

---

> ### Author Rebuttal · Authors · 2025-07-30
>
> We sincerely thank Reviewer ewdF for providing valuable feedback.
>
> >**Q1:** "The motivations could probably be discussed more explicitly." "why a quick reconstructor or even a flow-based model specifically needs to be used in this case"
>
> Sorry for the confusion. We would like to clarify our motivation further. In this work, our goal is to solve the **physics sensing task, which we define as a complete solution for understanding physical fields**. This task inherently involves two tightly coupled components: sensor placement and field reconstruction.
>
> If one focuses solely on field reconstruction, it suffices to train a reconstructor on given observations from fixed placements. If one focuses solely on sensor placement, it may even be done using simple heuristics or statistical priors, without the need for an explicit reconstructor.
>
> However, our aim goes beyond these decoupled formulations. We aim to enable data-driven physics sensing—given historical or pre-collected data, we want to determine not only where to measure, but also how to reconstruct from those measurements. Therefore, the **core contribution of our framework lies in the joint optimization of sensor placement and reconstruction**, which is critical for enabling practical and accurate sensing in physical environments.
>
> Practically, PhySense is designed to output both the **optimized placement and a reconstruction model**. This enables a **closed-loop physical sensing system**: once the sensors are deployed and real-world measurements are collected, the reconstruction model can immediately infer the full physical field based on these observations.
>
> We kindly remind this perspective may have been partially overlooked in your summary, which emphasized reconstruction but not the whole framework. And we will revise our introduction to better clarify this motivation.
>
> > **Q2:** "Why would a flow based model be required in this case?"
>
> Thank you for the thoughtful question. Below, we provide our detailed justification in four aspects.
>
> **(1) The task we tackle is inherently generative.**
>
> Inferring the full physical field from sparse observations is a highly underdetermined problem. In such scenarios, high-capacity generative models are well-suited. Notably, two of our strongest baselines, S3GM and DiffusionPDE, are also generative models. Moreover, compared to simple deterministic models trained with MSE, generative models are better equipped to avoid the overly smooth reconstructions often seen in ill-posed inverse problems, and thus provide superior physical fidelity.
>
> **(2) Flow models offer uncertainty quantification.**
>
> Flow-based models explicitly model the full conditional distribution, enabling uncertainty quantification. We reconstruct each input three times using different initial noises and then compute the standard deviation of the relative L2 error, as summarized in the table below. The results indicate a clear trend: as the number of sensors decreases, the conditional distribution becomes more diverse, leading to higher uncertainty (i.e., larger standard deviation). Further, we also calculate a pixel-wise variance map which reveals physically meaningful spatial patterns of uncertainty. However, due to NeurIPS’s rebuttal policy, we cannot include additional figures at this stage, but we promise to present pixel-wise uncertainty visualizations in the appendix.
>
> | Turbulent Flow                     | 200 sensors | 100 sensors | 50 sensors | 30 sensors |
> | - | - | - | - | - |
> | Standard deviation of relative L2 | 0.0043      | 0.0086      | 0.0210     | 0.0390     |
>
> **(3) Flow process enables progressive sensor placement optimization**
>
> As shown in $\underline{\text{Eq.2 in the main paper}}$, our sensor placement optimization is driven by the flow loss, which is formally defined in $\underline{\text{Eq.1}}$. A critical feature is that the flow model operates on a noise-data mixed input $\mathbf{X}_t$. In contrast, non-generative models cannot support such interpolated input. They typically start from either pure noise or a fixed initialization (e.g., zeros), and thus lack the ability to reflect the gradual transition from low to high fidelity representations. In our case, the flow process naturally induces **a progressive sensor placement optimization across different difficulty levels of reconstruction**, corresponding to different noise levels in the flow process, which yields a more informative and tractable feedback than non-generative models.
>
> To verify this insight, we conduct an ablation study where we remove this flow process feedback—**using pure noise instead of the noise-data mixture in the feedback loop**—and observe noticeable degradation in performance. This demonstrates that the flow-process feedback feature provides unique benefits for sensor optimization.
>
> | Turbulent Flow                                               | 200 sensors | 100 sensors | 50 sensors | 30 sensors |
> | - | - | - | - | - |
> | PhySense + random sensor placement                           | 0.1233      | 0.1527      | 0.2586     | 0.5176     |
> | PhySense + **noise-start** optimized sensor placement        | 0.1118      | 0.1293      | 0.1630     | 0.2286     |
> | PhySense + **mixed-start** optimized sensor placement (**our original submission**) | 0.1106  | 0.1257  | 0.1558 | 0.2157 |
>
> **(4) Flow models are efficient enough.**
>
> The number of training samples and the corresponding training time (in A100 GPU hours) for PhySense are reported in the table below. All baseline methods are trained using the **same dataset and number of epochs**, so the **overall training cost depends solely on model complexity, rather than whether the model is generative**. In this regard, PhySense exhibits a training cost comparable to other generative baselines. Moreover, PhySense is significantly more efficient at inference time, which maintains a relative L2 error below 0.2 with as few as 2 inference steps ($\underline{\text{Fig. 6c of main text}}$), significantly outperforming S3GM and DiffusionPDE.
>
> |                           | Turbulent Flow | Sea Temperature | Car Aerodynamics |
> | - | - | - | - |
> | Sample Number             | 10000          | 9843            | 75               |
> | Training Time (A100 hours) | 22             | 10              | 6                |
>
> > **Q3:** "how other point selection method would work in this problem and how those placements may affect the other reconstruction methods proposed"
>
> We conducted comprehensive experiments using requested placement baselines across Senseiver and PhySense under different sensor numbers on Turbulent Flow dataset. Specifically, we consider the following strategies:
>
> - **Random**: A general baseline with no prior knowledge.
> - **Grid**: We first place sensors on a uniform grid using the largest square number less than the budget, and then randomly distribute the remaining sensors. While this strategy ensures spatial coverage, it does not account for underlying physical dynamics. In practice, its performance is comparable to random placement, which is consistent with classical sampling theory [1], suggesting that regular sampling may not outperform randomized designs in complex data.
> - **Min-Max**: A heuristic that promotes spread-out sensor placements by iteratively placing the next sensor at the point that maximizes the minimum distance to existing sensors. However, this approach does not yield significant improvements over random placement in our experiments.
> - **Our Projected Gradient Descent**: An upper-bound variant that uses the corresponding reconstruction model (Senseiver or PhySense) and performs our projected gradient descent to optimize sensor locations.
> - **Optimized Sensor from PhySense**: To evaluate the generality of the learned placements by PhySense, we feed the placement into the *Senseiver*. **The resulting performance closely matches that of the PGD and significantly outperforms all other baselines**. This suggests that our learned placements effectively capture physically informative structures that generalize beyond the inductive biases of specific reconstruction models.
>
> | Turbulent Flow | Random | Grid   | Min-Max | Our Projected GD on the reconstruction model | Optimized Sensor from PhySense |
> | - | - | - | - | - | - |
> | Senseiver-200| 0.1842 | 0.1615 | 0.1549  | 0.1388| 0.1533|
> | Senseiver-100| 0.2316 | 0.2273 | 0.2091  | 0.1750 | 0.1833|
> | Senseiver-50| 0.3740 | 0.3845 | 0.3795  | 0.3025 | 0.3035 |
> | Senseiver-30| 0.5746 | 0.5869 | 0.5747  | 0.4570 | 0.4601|
> | PhySense-200| 0.1233 | 0.1235 | 0.1196  | 0.1106 | 0.1106|
> | PhySense-100| 0.1527 | 0.1521 | 0.1533  | 0.1257| 0.1257|
> | PhySense-50| 0.2586 | 0.2436 | 0.3157  | 0.1558| 0.1558|
> | PhySense-30| 0.5176 | 0.4930 | 0.5022  | 0.2157| 0.2157 |
>
> [1] Near-optimal signal recovery from random projections: Universal encoding strategies? IEEE TIT, 2006
>
> > **Q4:** "Are the selected sensor placements interpretable or have some characteristics in any way?  how sensitive are they to the GD process -- i.e., if I initialise my optimisation dfferently would the recovered sensor placements change?"
>
> First, the optimized sensor placements exhibit **a certain degree of physical interpretability**. For example, in $\underline{\text{Fig 5 of main text}}$ on the sea temperature dataset, the sensors tend to concentrate near coastal regions or ocean currents, where the physical dynamics are most active. This claim is further supported by all the showcases throughout the paper.
>
> Second, while varying the initial placements naturally leads to different final placements, the **resulting reconstruction performance remains consistently high** across these variants, shown in $\underline{\text{Sec 4.4 and Fig 6b}}$. This robustness suggests that PhySense does not tailor to a specific initialization but identifies informative placements across the solution space. It also provides insight that multiple runs may help explore the Pareto optimality of placements.

---

> > ### Comment · Reviewer_ewdF · 2025-08-04
> > **Response**
> >
> > I thank the reviewer for their response. Regarding the author's response with Q2, I am still somewhat unconvinced. Mainly, I am unsure if the task is really inherently generative (inverse problems have been studied long before diffusion models and similar, e.g., through experimental design works), and the remaining two points could also be solved by other methods as well (e.g., Bayesian methods can perform uncertainty quantification well). Some of these points could be better demonstrated by some comparison with those literature.
> >
> > However, (1) I still believe that generative models may provide some benefits especially in very complex problems and this is likely a matter of debate and further studies which this work may contribute to, and (2) I believe the remaining points are addressed well (particular some additional experiments on more naive benchmarks added in the rebuttals), and the work likely provides enough extension from current generative model methods to also involve physics bias. In light of this, I will increase the score for my evaluation, but also decrease the confidence for my review since I am not overly familiar with works in the generative model field.

---

> > > ### Author Response · Authors · 2025-08-05
> > >
> > > Thank you for your thoughtful feedback and for increasing the score. We appreciate your recognition of our contributions. Regarding the generative model discussion, as you mentioned, "generative models may provide some benefits, especially in very complex problems," and we agree this statement. Moreover, we believe that PhySense offers a practical solution for such complex tasks, where traditional methods may be less effective due to inherent complexity and non-linearity. Further, we appreciate your suggestions to compare with inverse problem methods and Bayesian approaches and will explore these comparisons in $\underline{\text{the future work section}}$ to further validate and refine our approach.

---

### Official Review · Reviewer_dimJ · 2025-06-29

**Clarity:** 4
**Significance:** 3
**Originality:** 3
**Rating:** 5
**Confidence:** 5

**Summary:**

The paper investigates physics‐sensing problems—reconstructing dense physical fields from sparse measurements—by jointly learning (i) a flow-matching generative reconstructor conditioned on arbitrary sensor sets and (ii) a geometry-aware projected-gradient algorithm that relocates sensors to near-A-optimal positions. A key theoretical result shows that minimizing the “flow loss” used in placement optimization yields a two-sided quadratic bound on classical A-optimality  along with a theorem that establishes unbiasedness of the reconstructor. Experiments on three benchmarks—channel turbulence, global sea-surface temperature, and 3-D car aerodynamics—report up to 62 % lower reconstruction error than prior art and reveal non-obvious optimal sensor layouts.

**Questions:**

How critical is the Conditional-Gaussian Assumption for optimality? Perhaps take a toy problem and evaluate it?

Evaluate robustness or extend analysis to sub-Gaussian / heavy-tailed priors; provide empirical test with non-Gaussian toy problem?

Average absolute gradient of the reconstruction error w.r.t. each potential location, overlaid with final sensor positions. This could reveal whether the optimiser gravitates to high-information or high-variance locations.

**Ethical Concerns:**

["NO or VERY MINOR ethics concerns only"]

**Final Justification:**

Edit: After reviewing the response, and considering what is possible between today and the revision deadline, I am changing my decision from borderline accept to accept.

**Limitations:**

Yes

**Quality:**

3

**Strengths And Weaknesses:**

Strengths:
- Coupling flow-matching reconstruction with a placement optimizer that is differentiable and geometry-constrained is novel & elegant.
- The polynomial control between flow loss and A-optimality provides a formal link missing in some other (but not all!) sensor-placement works
- Evals are comprehensive: 3 diverse domains (regular grid, land–sea mask, unstructured 3-D mesh) demonstrate both generality and significant gains over five baselines, including strong generative models
- Ablation and diagnostics are good, attention visualizations build reader confidence in the design choices
- the optimized 15-sensor sea-temperature setup achieves accuracy comparable to 100 random sensors, hinting at real cost savings in field deployments

Weaknesses
- While the manuscript demonstrates strong empirical gains, the chain that links the algorithm’s design choices to those gains is still opaque. I am unsure why it outperforms both classical A-optimal design and recent deep surrogates.
- Theorem 3.5 proves a quadratic relation to A-optimality in expectation, but the paper never shows that low flow-loss actually correlates with low test MSE in practice.
- The benchmarks are legitimate, but I have seen better results on these benchmarks with other classical methods (Even with QDEIM, which is a lightning fast method with good bounds)
- Where is the physics in the method? It is being applied to fields... The `physics' itself is not utilized. The model would behave identically if the snapshots came from video frames, as long as the tensors have the same shape.

---

> ### Author Rebuttal · Authors · 2025-07-30
>
> We sincerely thank Reviewer dimJ for providing insightful suggestions.
>
> > "Q1: the chain that links the algorithm’s design choices to those gains is still opaque. I am unsure why it outperforms both classical A-optimal design and recent deep surrogates"
>
> First, we clarify a possible misunderstanding: we do not claim to outperform A-optimal design empirically. Rather, our theoretical analysis proves a bidirectional polynomial control relationship between the flow loss and the A-optimality objective in $\underline{\text{Theorem 3.5}}$. Importantly, A-optimal design focuses solely on sensor placement and does not support field reconstruction, while PhySense addresses the both. Thus they are not directly comparable.
>
> Below, we provide our detailed justification for the strong empirical gains in three aspects.
>
> **(1) Flow reconstructor has better model capacity.**
>
> Inferring the full physical field from sparse observations is a highly underdetermined problem. In such scenarios, high-capacity generative models are well-suited. Moreover, compared to those deterministic models trained with MSE, generative models are better equipped to avoid the overly smooth reconstructions often seen in ill-posed inverse problems, and thus provide superior physical fidelity.
>
> **(2) Flow process enables progressive sensor placement optimization.**
>
> As shown in $\underline{\text{Eq.2}}$, our sensor placement optimization is driven by the **flow loss**, formally defined in $\underline{\text{Eq.1}}$. A critical feature is that the flow model operates on a **noise-data mixed input $\mathbf{X}_t$**. In contrast, non-generative models cannot support such interpolated input. They typically start from **either pure noise or a fixed initialization** (e.g., zeros), and thus lack the ability to reflect the gradual transition from low to high fidelity representations. In our case, the flow process naturally induces **a progressive sensor placement optimization across different difficulty levels of reconstruction**, corresponding to different noise levels in the flow process, which yields a more informative and tractable feedback than non-generative models.
>
> To verify this insight, we conduct an ablation study where we remove this flow process feedback—**using pure noise instead of the noise-data mixture in the feedback loop**—and observe noticeable degradation in performance. This demonstrates that the flow-process feedback feature provides unique benefits for sensor optimization.
>
> | Turbulent Flow | 200 sensors | 100 sensors | 50 sensors | 30 sensors |
> | - | - | - | - | - |
> | PhySense + random sensor placement| 0.1233      | 0.1527      | 0.2586     | 0.5176     |
> | PhySense + **noise-start** optimized sensor placement| 0.1118      | 0.1293      | 0.1630     | 0.2286     |
> | PhySense + **mixed-start** optimized sensor placement (**origin**) | 0.1106      | 0.1257      | 0.1558     | 0.2157     |
>
> **(3) Senseiver under optimized sensor placement is still worse.**
>
> We further compare a stronger baseline—**Senseiver with sensor placement optimized by our project gradient descent**—on the turbulent flow. Despite optimized placement, Senseiver still underperforms PhySense, especially when the sensor number is low. This highlights that while placement is important, model capacity remains a limiting factor.
>
> | Turbulent Flow | 200 sensors | 100 sensors | 50 sensors | 30 sensors |
> | - | - | - | - | - |
> | Senseiver + random sensor placement    | 0.1842      | 0.2316      | 0.3740     | 0.5746     |
> | Senseiver + optimized sensor placement **using our PGD** | 0.1388      | 0.1750      | 0.3025     | 0.4570     |
> | PhySense + random sensor placement     | 0.1233      | 0.1527      | 0.2586     | 0.5176     |
> | PhySense + optimized sensor placement  | 0.1106      | 0.1257      | 0.1558     | 0.2157     |
>
> > "Q2: Theorem 3.5 proves a quadratic relation to A-optimality in expectation, but the paper never shows that low flow-loss actually correlates with low test MSE in practice."
>
> This is a well-acknowledged property in the field of generative models. The flow loss $\mathcal{L}_{\text{flow}}$ quantifies the deviation of the learned velocity field from the ideal transport direction, conditioned on sparse sensor observations. A lower flow loss implies that the model has accurately learned how to map noise to the target data manifold under the given sensor conditions. As a result, samples generated under low flow loss are more likely to align with the true conditional distribution, leading to reconstructions that are more physically plausible and have lower test MSE.
>
> >"Q3: I have seen better results on these benchmarks with other classical methods (Even with QDEIM, which is a lightning fast method with good bounds)"
>
> Thank you for this point. To the best of our knowledge, we are not aware of classical methods demonstrating better results on these complex benchmarks. QDEIM and the SSPOR we compare against share a similar idea: they both leverage POD (Proper Orthogonal Decomposition) to reduce the problem dimensionality and select sensor locations based on the dominant subspace. However, SSPOR shows limited performance in our experiments. If you may have suggestions for additional baselines, we would be happy to include them in the discussion stage.
>
> Moreover, **deep learning methods exhibit clear advantages in highly nonlinear and high-dimensional settings**. These scenarios often involve intricate spatial correlations and nontrivial patterns that classical linear subspace methods may fail to capture. Further, deep models benefit significantly from scaling up data and model capacity—just as we have seen with large language models like ChatGPT—indicating a strong potential for performance gains. Therefore, **we believe investigating deep learning–based physics sensing remains both valuable and promising**, especially for real-world applications.
>
> >"Q4: Where is the physics in the method? The model would behave identically if the snapshots came from video frames, as long as the tensors have the same shape"
>
> While we do not impose explicit physical constraints during model training, the data used in our setting already reflects underlying physical laws and thus **implicitly lies on a low-dimensional physical manifold**. This inductive structure is what makes the reconstruction process—from sparse to dense observations—both feasible and effective in physical domains. In contrast, such low-dimensional structure is generally absent in vision datasets: it is not feasible to recover a video from a few individual pixel sequences, nor to accurately interpolate intermediate frames based solely on distant snapshots. While the tensor shape ensures compatibility with deep models, it is ultimately the data distribution that determines whether a method can work.
>
> Further, we agree that incorporating physics explicitly can be beneficial. Since it is orthogonal to our current focus, our framework is complementary to such efforts and can potentially integrate with them in future work.
>
> > "Q5: How critical is the Conditional-Gaussian Assumption for optimality? Evaluate robustness or extend analysis to sub-Gaussian / heavy-tailed priors"
>
> The **Gaussian assumption is commonly used in theoretical formulations due to its analytical tractability**, particularly the ability to derive closed-form expressions for conditional distributions. This modeling choice has been widely adopted in prior works on sensor placement and experimental design [1]. Extending the theoretical analysis to sub-Gaussian or heavy-tailed priors is certainly a valuable direction; however, removing the Gaussian assumption significantly poses substantial technical challenges. We sincerely appreciate your suggestion and will consider this as an important future work.
>
> Importantly, we emphasize that this assumption is used **solely for theoretical analysis**—our method **does not impose any distributional assumptions during training or evaluation**. In fact, the empirical distributions encountered in our experiments are often more complex than both sub-Gaussian and heavy-tailed settings. The strong and consistent empirical performance across a wide range of datasets provides further evidence of the robustness and practical effectiveness of our approach.
>
> [1] *Near-Optimal Sensor Placements in Gaussian Processes: Theory, Efficient Algorithms and Empirical Studies*, JMLR 2008
>
> > "Q6: Average absolute gradient of the reconstruction error w.r.t. each potential location, overlaid with final sensor positions. This could reveal whether the optimiser gravitates to high-information or high-variance locations."
>
> Many thanks for the insightful analysis experiment. We observe that high-variance regions tend to form continuous, band-like structures, while high-information points—measured by the gradient of reconstruction error with respect to each location—are often more discrete and spatially separated. Interestingly, **a large part of the selected sensors fall within the high-variance regions**. And **every selected sensor is found near a region of high information points, and their spatial distributions have similar patterns**. This observation suggests that the optimizer is not merely gravitating toward regions of maximal variance but is instead strategically identifying spatially separated, high-sensitivity locations, which better capture the global dynamics of the physical field. From a sensing perspective, this is quite reasonable: **maximizing overall information gain often requires sampling diverse and non-redundant locations, rather than densely covering localized variance clusters**. Due to the NeurIPS rebuttal policy, we are unable to include additional figures at this stage. However, we promise to include visualizations comparing high-information points, high-variance regions, and the final optimized placements in the appendix.

---

> > ### Author Response · Authors · 2025-08-07
> >
> > Thank you again for the detailed review on our paper. With **only two days** remaining in the discussion period, we would greatly appreciate knowing whether our rebuttal has satisfactorily addressed your concerns or whether there are any remaining issues we should clarify.
> >
> > Please let us know if additional information would be helpful. We are happy to provide further details.

---

> > ### Comment · Reviewer_dimJ · 2025-08-08
> > **Thank you for the revisions.**
> >
> > Thank you for performing more detailed evaluations. I am happy to change the decision to accept.

---

> > > ### Author Response · Authors · 2025-08-09
> > >
> > > Thank you again for your detailed review and thoughtful replies. Your willingness to raise the score is very meaningful to us. Besides, as a gentle reminder, the discussion period ends in **less than 12 hours**; if the score is not updated before the deadline, your assessment may not be fully reflected to the AC. We sincerely appreciate your time and consideration.

---

### Official Review · Reviewer_X4JQ · 2025-07-01

**Clarity:** 3
**Significance:** 3
**Originality:** 3
**Rating:** 5
**Confidence:** 2

**Summary:**

The paper proposes PhySense, a two-stage approach that both reconstructs a physical field from sparse sensing data, as well as optimizes the sensor locations to be optimal. The paper first uses a flow-based generative model to generate the physical field conditioning on a sparse set of measurement data. This learns a conditional generative model given input of different numbers and locations of the sensors. Then, the learned generator is used to refine the set of sensors' locations. The paper provides both theoretical and experimental guarantees about the convergence and performance gain compared to existing models.

**Questions:**

1. How is the flow loss calculated in Eq.(2)? I thought the optimization is performed during test time and the ground truth physical field (i.e., $X_1$) is not available.
2. Assumption 3.2: Is this assumption reasonable? Why would it be reasonable to assume that the target physical field follows a zero-mean Gaussian distribution?
3. The paper only evaluates the L2 loss of the modeled field. Is this a reasonable metric? I would imagine that the reconstructed physical field would be used for downstream applications that involve perhaps other quantities from the field (e.g., gradients, Hessians).

**Ethical Concerns:**

["NO or VERY MINOR ethics concerns only"]

**Final Justification:**

After rebuttal, I am clear of my original questions and had a better understand of the paper. I believe the authors have done a thorough job in terms of theoretical grounding and experimentally validating their proposed joint sense + reconstruction pipeline. I think this joint learning approach enables a wide range of applications and future directions for research. Therefore, I raised my score to accept.

**Limitations:**

Yes

**Quality:**

3

**Strengths And Weaknesses:**

## Strengths
1. The paper has a strong theoretical section in which it proves some theoretical guarantees of their method's convergence. These theoretical analysis boosts the credibility of the paper's method and their experimental performance.
2. Compared to previous works that only focus on reconstructing the physical field, the paper proposes a novel joint optimization of both the physical field and the sensor locations, showing that a joint optimization of both fronts would lead to better performance.
3. The experimental section contains multiple settings covering a wide range of physical phenomena. The proposed method achieves non-trivial improvements over the existing methods in all these settings, demonstrating the effectiveness of their approach.

## Weaknesses
1. From reading the paper, it seems that no physical constraints are used when learning the generative model. Would adding some sort of physically informed loss would help with the reconstruction?
2. I found it hard to motivate the physical meaning of the optimized sensor locations in the second stage of the method. The sensor location optimization seems to be finding the optimal sensor configuration based on the learned generative model, so I wonder if there is some kind of overfitting happening when doing the second stage optimization. For example, would the physical field reconstructed using other methods with the optimized sensor locations improve?

---

> ### Author Rebuttal · Authors · 2025-07-30
>
> Many thanks to Reviewer X4JQ for providing a detailed review and insightful questions.
>
> > **Q1:** "no physical constraints are used when learning the generative model. Would adding some sort of physically informed loss would help with the reconstruction"
>
> First, we would like to emphasize that reconstruction is only one part of our framework—our primary contribution lies in the **joint optimization of sensor placement and reconstruction**, which is critical for accurate physics sensing.
>
> While we do not impose explicit physical constraints during model training, the data used in our setting already reflects underlying physical laws and thus **implicitly lies on a low-dimensional physical manifold**. This inductive structure, which is often absent in general vision datasets, is what makes the reconstruction process—from sparse to dense observations—both feasible and effective in physical domains.
>
> Moreover, in our target scenarios, we are **not aiming to reconstruct a fully closed physical system governed by complete equations** (common in real settings). For example, we focus on reconstructing the temperature of ocean surface rather than the full 3D ocean dynamics, or the surface flow over a car rather than the complete surrounding velocity field. **In such partially observed systems, it is often not feasible to impose full equation-based constraints**, as many physical variables are unobserved or unmeasurable. This limitation applies equally to all baseline methods we compare against.
>
> Further, we agree that incorporating physically informed losses can be beneficial. While it is orthogonal to our current focus, our framework is complementary to such efforts and can potentially integrate with them in future work.
>
> > **Q2:** "the physical meaning of the optimized sensor locations in the second stage of the method. The sensor location optimization seems to be finding the optimal sensor configuration based on the learned generative model, would the physical field reconstructed using other methods with the optimized sensor locations improve"
>
> First, the optimized sensor placements learned in the second stage of our method exhibit **a certain degree of physical interpretability**. For example, as shown in Figure 5 on the sea temperature dataset, the selected sensors tend to concentrate near **coastal regions or ocean currents**, where the physical dynamics are most active and spatial variability is high.
>
> Moreover, we do not view the sensor optimization as overfitting to our reconstruction model. Rather, our framework **jointly learns**:
>
> 1. The **global biases of the underlying physical field**, which are **model-agnostic** and thus transferable across reconstruction models.
> 2. The **model-specific inductive biases**, which allow our flow model to extract more information from sparse observations.
>
> To further investigate this, as per your request, we conducted an ablation study comparing:
>
> - **Senseiver + random sensor placement**
> - **Senseiver + sensor placement optimized by Senseiver using our PGD**
> - **Senseiver + sensor placement optimized by our PhySense**
>
> Interestingly, the third setting—Senseiver using sensor locations optimized by PhySense—yields **significantly better performance** than random placement, suggesting that our learned placement captures **physically informative structures** that go beyond the reconstruction model's own inductive bias. The quantitative results are provided in the following table.
>
> | Turbulent Flow                                               | 200 sensors | 100 sensors | 50 sensors | 30 sensors |
> | ------------------------------------------------------------ | ----------- | ----------- | ---------- | ---------- |
> | Senseiver + random sensor placement                          | 0.1842      | 0.2316      | 0.3740     | 0.5746     |
> | Senseiver + sensor placement optimized by Senseiver using our PGD | 0.1388      | 0.1750      | 0.3025     | 0.4570     |
> | Senseiver + sensor placement optimized by our PhySense       | 0.1533      | 0.1833      | 0.3035     | 0.4601     |
> | PhySense + optimized sensor placement                        | 0.1106      | 0.1257      | 0.1558     | 0.2157     |
>
> > **Q3:** "How is the flow loss calculated in Eq.(2)? I thought the optimization is performed during test time and the ground truth physical field (i.e., ) is not available."
>
> The optimization in Eq. (2) is not performed at test time. Rather, **it is part of the second stage of training, where ground-truth physical fields are fully available**. For instance, we utilize 27 years of global ocean reanalysis data with randomly sampled sensor placements to train a reconstruction model. This model then provides feedback to guide the optimization of sensor placement via projected gradient descent. **The two steps occur entirely during training time.**
>
> During actual deployment (inference time), only the optimized sensor locations are used to collect real measurements, which are then passed through the trained reconstruction model to recover the full field. **Performing sensor optimization per test case is impractical**, as it would require access to ground-truth fields unavailable in real world. Instead, our framework produces a **consistent, scene-specific sensor placement that generalizes across test cases and can be deployed in practice**.
>
> > **Q4:** "Why would it be reasonable to assume that the target physical field follows a zero-mean Gaussian distribution"
>
> The assumption that the mean is zero in Assumption 3.2 is adopted primarily for **notational simplicity**, which is a standard practice especially when the data has been normalized. Importantly, our theoretical analysis focuses exclusively on the **covariance structure** of the field (i.e., the variance components), and the specific form of the mean does not influence the derivations or conclusions. As such, the results remain valid regardless of the centering.
>
> Moreover, the **Gaussian assumption is commonly used in theoretical formulations due to its analytical tractability**, particularly the ability to derive closed-form expressions for conditional distributions. This modeling choice has been widely adopted in prior works on sensor placement and experimental design [1].
>
> Furthermore, we emphasize that this assumption is made solely for theoretical analysis—we do not impose any such distributional assumption during training or evaluation. The strong empirical performance of our method across diverse datasets further validates its practical effectiveness.
>
> [1] *Near-Optimal Sensor Placements in Gaussian Processes: Theory, Efficient Algorithms and Empirical Studies*, JMLR 2008
>
> > **Q5:**  "The paper only evaluates the L2 loss of the modeled field. Is this a reasonable metric? other quantities from the field (e.g., gradients, Hessians)"
>
> Thank you for the thoughtful question. We agree that evaluation metrics should reflect meaningful physical fidelity. In our experiments, we primarily report **relative L2 loss**, which is widely used and provides an **intuitive and scale-invariant measure of reconstruction accuracy**. Compared to absolute L2 loss, the relative version avoids being dominated by the scale of the underlying physical field, which can vary significantly across data.
>
> Further, as per your request, we additionally evaluate the **relative L2 error of the gradient fields** on sea temperature. As shown in the table below, our method achieves significantly lower gradient error compared to the second-best model, Senseiver, demonstrating its superior physical fidelity.
>
> | Sea Temperature (relative L2 of gradient field) | 100 sensors | 50 sensors | 25 sensors | 15 sensors |
> | ----------------------------------------------- | ----------- | ---------- | ---------- | ---------- |
> | Senseiver + random sensor placement             | 0.5456      | 0.5470     | 0.5501     | 0.5515     |
> | PhySense + random sensor placement              | 0.4877      | 0.4884     | 0.4890     | 0.4892     |
> | Relative promotion                              | 11.87%      | 11.98%     | 12.51%     | 12.74%     |

---

> > ### Comment · Reviewer_X4JQ · 2025-08-05
> >
> > Thank the authors for the thorough responses. Most of my questions have been addressed, and I think I understand the paper better. I will raise my score. It might be interesting to extend the work to indirect measurements of the reconstruction field (e.g., 2D images for 3D scenes).

---

### Official Review · Reviewer_LEF1 · 2025-07-03

**Clarity:** 3
**Significance:** 4
**Originality:** 4
**Rating:** 5
**Confidence:** 4

**Summary:**

PhySense proposes a two-stage reconstruction model that uses an efficient flow based generative model to infer a reconstructed physical field conditioned on a set of sparse pointwise sensor measurements, then updates sensor positions by performing a gradient step that improves reconstruction accuracy and then projects sensor positions back to the domain (e.g. any set the sensors are restricted to). This process is repeated for a fixed number of rounds and –as the paper proves–is guaranteed to produce the A-optimal (e.g. variance minimizing) set of sensor positions. The novel use of flow matching in this setting already leads to improved one-shot reconstructions of physical fields over a range of prior works and the additional optimized sensor placement further improves performance. The framework is general enough that–as the paper demonstrates in experiments–the particular architecture of the velocity field can be interchanged depending on the specific application.

**Questions:**

* Does PhysSense take additional measurements at each step of projected gradient descent as its optimizing sensor positions? It's not clear to me whether the sensor optimization is a performed with real measurements or generate data.
* Where does this method stand in relation to approaches like Thompson sampling? Specifically, I'm curious if PhySense can be applied to adaptive sampling problems, where "sensors" are not so much a physical device that is being positioned but rather a position at which a measurement is collected (e.g. a camera moving around an object)?
* On line 191 is the conditional random variable supposed to be centered on 0 or X1(p)?

**Ethical Concerns:**

["NO or VERY MINOR ethics concerns only"]

**Final Justification:**

The paper introduces a clear, well-motivated two-stage framework for reconstructing physical fields from sparse sensors and optimizing sensor placement, with theory guaranteeing optimality. My concern about whether optimization required new measurements was resolved in the rebuttal, which clarified it is done offline with historical data. With this, I view the work as technically solid, clearly written, and impactful, and maintain my strong acceptance recommendation.

**Limitations:**

Yes.

**Quality:**

4

**Strengths And Weaknesses:**

strengths:
1. The presentation is exceedingly clear, providing excellent background and intuition on the three core areas leveraged in the method: sparse reconstruction, optimized sensor placement, and flow-based matching. I found the method section very easy to follow as a result, despite the fact that it incorporates many disparate subfields.

2. The method is straightforward, yet relies on an elegant theory proving that A-optimality is obtained through performing gradient descent on the flow residual with respect to the sensor positions. This raises many interesting follow up questions around how to achieve alternative sensor placement objectives–some of which were summarized earlier in the paper–and the interplay these objectives have on reconstruction methods.

weaknesses

1. The comparison against other reconstruction methods is partially unclear to me. It seems like in the process of optimizing sensor position, PhysSense makes additional measurements and this leads to more effective sensor positions being utilized than in the naive case–even if an equal number of measurements are directly applied to reconstruction. This could also be a misunderstanding on my part though, either in how PhySense is utilizing its sample budget or whether the sensor optimization is run as a pre-processing step using only the domain geometry and sampled PDE solutions.

---

> ### Author Rebuttal · Authors · 2025-07-30
>
> We would like to sincerely thank Reviewer LEF1 for providing a detailed review with positive evaluations and insightful questions.
>
> >**Q1:** "how to achieve alternative sensor placement objectives and  the interplay these objectives have on reconstruction methods"
>
> In our paper, we rigorously prove in $\underline{\text{Theorem 3.5}}$ that the flow-loss minimization objective polynomially controls the classical A-optimal criterion. Furthermore, under $\underline{\text{Assumption 3.2}}$, **this mutual control relationship naturally extends to both D-optimality and E-optimality objectives**. Specifically, their formulations can be simplified to
>
> $\mathcal{L}\_D$(p) := det($\Sigma\_p$) = $\Pi_{i=1}^d$ $λ_i$(p)
> $\mathcal{L}_E$(p) := $λ\_{max}$($\Sigma\_p$)
>
>
>
> It is evident that both $\mathcal{L}_D$ and $\mathcal{L}_E$ can be polynomially bounded above and below by $\mathcal{L}_A$ due to standard spectral inequalities. Consequently, through the control of $\mathcal{L}_A$, the flow-loss objective also polynomially controls $\mathcal{L}_D$ and $\mathcal{L}_E$ in both directions. Therefore, the flow-loss objective serves as a spectrum-aware surrogate that can also guide D-optimal and E-optimal sensor placement with bounded distortion.
>
> We will include this new finding in the main text and provide detailed proofs in the appendix to further demonstrate the generality and extensibility of our framework.
>
> >**Q2:** "The comparison against other reconstruction methods is partially unclear to me."  "How PhySense is utilizing its sample budget." " whether the sensor optimization is performed with real measurements or generate data"
>
> PhySense can be viewed as a pre-processing stage that leverages historical or pre-collected data. For instance, we utilize 27 years of global ocean reanalysis data with randomly sampled sensor placements to train a reconstruction model. This model then provides feedback to guide the optimization of sensor placement via projected gradient descent. In this way, PhySense jointly optimizes both the sensor placement and the reconstruction model with respect to the data distribution. **During actual deployment (inference time), only the optimized sensors are used to collect real measurements—thus, the sample budget strictly equals the number of deployed sensors.** These measurements are then passed through the trained reconstruction model to reconstruct the full physical field.
>
> Jointly optimizing sensor placement and reconstruction is a key contribution of our work. **To isolate and evaluate the performance of the reconstruction models alone, we adopt a randomized sensor placement strategy as an idealized setting**. This allows us to assess reconstruction quality independent of the placement method. All baseline methods follow the same pipeline for a fair comparison across all tables when the placement strategy is set to random.
>
> > **Q3:** "PhySense stand in relation to adaptive sampling approaches like Thompson sampling"
>
> Thank you for the insightful question. Although our current work focuses on static sensor placement, it is closely related in spirit to adaptive sampling strategies such as Thompson Sampling, which also aim to select informative locations under uncertainty. Given the importance and practical relevance of adaptive sensing, we view this as a promising direction for future work.
>
> > **Q4:** "On line 191 is the conditional random variable supposed to be centered on 0 or X1(p)"
>
> The conditional Gaussian assumption in $\underline{\text{Assumption 3.2}}$ models the target distribution as centered at 0 **for notational simplicity**, which is a standard treatment especially when the **data has been normalized**. Indeed, in practice, the observed values at the sensor locations $\mathbf{X}_1(\mathbf{p})$ are fixed (i.e., conditioned on), and the conditional variance at $\mathbf{X}_1(\mathbf{p})$ is zero. Therefore, one could equivalently center the distribution around $\mathbf{X}_1(\mathbf{p})$ instead of 0. Furthermore, since **our theoretical analysis focuses solely on the covariance structure (i.e., the variance components), the specific form of the mean does not affect the results**. Thus, the derivations remain valid regardless of the centering chosen.

---

> > ### Comment · Reviewer_LEF1 · 2025-08-05
> >
> > Thank you for the clarifications. The rebuttal addresses my primary confusion; I now understand that in Eq. (2) the gradient of the flow loss can be computed over historical or pre-collected data. I therefore maintain my recommendation for acceptance.

---

### Note · Authors · 2025-08-13

Dear Area Chair and Reviewers,

Thank you for your engagement and valuable feedback. We hope that we have addressed your concerns during the rebuttal and discussion stages. Here we want to summarize and emphasize some key aspects:

**Our contribution:** We propose PhySense, a synergistic two-stage framework that learns to jointly reconstruct physical fields and to optimize sensor placements with theoretical guarantees. It achieves consistent state-of-the-art reconstruction accuracy with 49% relative gain and discovers informative sensor placements.

This paper has received consistent positive reviews from 4 reviewers. Especially, the novelty and contributions of PhySense are appreciated by all the reviewers:

* LEF1: *“**The presentation is exceedingly clear**.”* *“Relies on an **elegant theory proving**.”*
* X4JQ: *“The paper has a strong theoretical section.”* *“The paper proposes a **novel joint optimization**.”* *“The proposed method achieves **non-trivial improvements**.”*
* dimJ: *“The proposed method is **novel and elegant**.”* *“The theory **provides a formal link**”* *“**Hinting at real cost savings in field deployments**.”*
* ewdF: *“**Has real use case**.”* *“**The proposed solution is interesting**.”* *“**Theoretical results are sufficient for the claims**.”*

**Summary of Rebuttal:** During rebuttal, following the insightful questions from reviewers, we further improved this work in the following aspects:
* LEF1: Clarified the full pipeline for sensor placement optimization.
* X4JQ: Added a gradient-based metric to further demonstrate physical fidelity.
* ewdF: Clarified the overall task motivation and design choices.
* dimJ & ewdF: Provided an ablation removing the flow-process feedback, showing the its advantage for sensor placement optimization.
* X4JQ, dimJ & ewdF: Added more placement baselines, further demonstrating the advantage of our PGD strategy, and showed that PhySense-optimized sensors improve other reconstructors (e.g., Senseiver), evidencing cross-model transfer.

We thank all reviewers and the AC for their careful assessments and constructive discussion, which has helped us significantly in improving our work. In particular, we appreciate the **score increases from X4JQ, dimJ, and ewdF**, reflecting further recognition of the work.

We hope the clarifications, additional analyses, and the strong empirical evidence collectively demonstrate the merit and impact of PhySense.

We thank you again for your time and consideration.

---

### Decision · Program_Chairs · 2025-09-17

**Decision:**

Accept (oral)

**Comment:**

All reviewers acknowledge the excellent presentation and theotretical background. They state that:

LEF1:  The paper is "technically solid, clearly written, and impactful"

dimJ: "Coupling flow-matching reconstruction with a placement optimizer that is differentiable and geometry-constrained is novel & elegant"  and "Evals are comprehensive: 3 diverse domains (regular grid, land–sea mask, unstructured 3-D mesh) demonstrate both generality and significant gains over five baselines, including strong generative models"

X4JQ "The experimental section contains multiple settings covering a wide range of physical phenomena. The proposed method achieves non-trivial improvements over the existing methods in all these settings, demonstrating the effectiveness of their approach."

All above reviewers provide a strong accept. The fourth reviewer provides a weak accept, but mentions they are not confident in their rating, as they are not from the research field.

Given the above reviews AC clearly recommends to accept the paper. As it presents a technical novel and theoretically proven approach that yields a gain of 49%, the AC believes this paper is of significant importance to the community and is likely to make a broad impact.